# Can Saharan dust deposition impact snowpack stability in the French Alps?

Oscar Dick[1,*], Léo Viallon-Galinier[1,2,*], François Tuzet[1,*], Pascal Hagenmuller[1], Mathieu Fructus[1], Benjamin Reuter[1], Matthieu Lafaysse[1], and Marie Dumont[1]

[1]Univ. Grenoble Alpes, Université de Toulouse, Météo-France, CNRS, CNRM, Centre d'Etudes de la Neige, 38000 Grenoble, France
[2]Ecole des Ponts, Champs-sur-Marne, France
[*]equal contribution

**Correspondence:** Léo Viallon-Galinier (leo.viallon@meteo.fr)

**Abstract.** Saharan dust deposits can turn snow covered mountains into a spectacular orange landscape. When avalanches release, a formerly buried dust layer can become apparent, possibly marking the base of the crown. This appearance may suggest a relation between avalanche release and the prior deposited dust, which found mention among recreationists and avalanche professionals alike. While dust deposition affects the absorption of solar energy altering snowpack temperatures and melt rates, to date, there is no clear scientific evidence that dust deposition can significantly modify avalanche activity. Here we investigate, using an ensemble snow cover model, the impact of dust deposition on snow properties and mechanical stability by comparing simulations with and without dust deposition for synthetic and observed dust deposition events. The study focuses on two typical avalanche situations: artificial triggering on persistent weak layers and natural release of wet-snow avalanches. We study several situations with and without dust deposition and demonstrate how sensitive the impact of dust deposition is to the deposited dust mass, the slope aspect, the elevation and the meteorological conditions following the dust deposition. The additional energy absorbed by the dust layer speeds up warming and may advance surface wetting to ease the formation of a melt-freeze crust. If the crust is buried, the phenomenon of a strong temperature gradient close to the crust may promote the formation of persistent weak layers inside the snowpack after weak layer burial. On the other hand, the melt-freeze crust may also lead to an increase of snowpack stability by redistributing the stress applied to buried weak layers. Regarding wet-snow instabilities, we show that dust deposition can advance the onset of wet-snow avalanche activity by up to one month in spring, as hypothesized in previous studies. Thus, the impact of Saharan dust deposition on snow mechanical stability can be either neutral, positive or negative, depending on the local snow and meteorological conditions. Even though not all physical processes are implemented, state-of the art snow cover models are able to mimic the speed-up of crust formation and snow instability models can point out relevant situations for avalanche forecasting.

## 1 Introduction

Snow avalanches are a major natural hazard, threatening infrastructures and human life in mountain areas throughout the world (Schweizer et al., 2021). Despite major breakthroughs in the understanding of the release processes since the end of the twentieth century (Schweizer et al., 2016), the impact of mineral dust deposition (hereafter referred as dust) on snowpack stability is still poorly understood. The lack of a clear message from the snow science community, has left room for interpretation of fallen avalanches with dust in the bed and so, the presence of a dust layer was often associated with a decrease of snowpack stability. Dust can indeed be regularly observed at the bed surface of triggered avalanches in affected regions following dust outbreaks. In April 2016, for instance, a French skier died caught in an avalanche showing a dust layer on the bed surface (Chomette et al., 2016). These accidents and the lack of understanding the role of dust in snow instability are the main motivation of the present study.

In Europe, according to LIDAR and satellite observations, dust clouds mainly come from the Saharan desert and can be carried up to Norway (Knippertz and Todd, 2012) and Greenland (Francis et al., 2018). The convection forces, due to the strong heating of the Sahara and the Sahel regions, cause an uplifting of huge quantities of dust. A significant part, estimated as 80-120 Gt per year, is transported northward across the Mediterranean sea and then dropped off on the European continent, including the Alps (Barkan et al., 2004, 2005). Saharan dust outbreaks are sporadic events generally occurring from April to August in the European Alps (Greilinger and Kasper-Giebl (2021) and references therein). However major outbreaks can also occur during the winter as the one recently observed in the Western Alps on the sixth of February 2021 (Réveillet et al., 2021). In the Caucasus mountains, these strong winter-time Saharan dust outbreaks have even been reported to become more and more frequent, hypothetically due to the polar amplification (Varga, 2020).

Mineral dust is one of the many light absorbing particles that can be deposited on snow covered surfaces (Skiles and Painter, 2018). These light absorbing particles lower the albedo of snow surfaces where they are deposited and increase the amount of absorbed solar energy in the visible wavelengths (Wiscombe and Warren, 1980). Even a minor dust deposition can reduce albedo by a few percent and cause surface melting in daylight hours, despite relatively weak solar radiation (Landry, 2014). The accelerated metamorphism, caused by the additional energy absorption, further leads to an albedo decrease in the visible and near-infrared domain. This causes a positive feedback loop on the absorbed solar energy, further accelerating surface melting (Hansen and Nazarenko, 2003; Painter et al., 2007; Skiles and Painter, 2019, 2018). Besides, the chemical composition of light absorbing particles at the snow surface may also impact the snow mechanical properties provided a high dust concentration (Hammonds and Baker, 2016; Meinander et al., 2014). However, these mechanisms are still poorly understood and we only focus, in the following, on the radiative impact of mineral dust. Various types of light absorbing particles can be found in the alpine snowpack such as black and organic carbon (Hadley and Kirchstetter, 2012), volcanic ashes (Liu et al., 2014) or snow algae (Remias et al., 2010). In the European Alps, however, Saharan mineral dust has been hypothesized to

be a driver of avalanche activity due to the sporadic but intense nature of the depositions (Chomette et al., 2016). That is why we only focus in the present study on Saharan mineral dust impact on snowpack stability.

Up to date, few studies have investigated the impact of dust outbreaks on snowpack stability. The Center for Snow and Avalanche Studies in Silverton, Colorado, has documented and monitored 91 dust events between 2005 and 2014 at the Senator Beck Basin Study Area at Red Mountain Pass in the San Juan Mountains, and at ten other locations in the Colorado mountains (Landry, 2014). This study presents a complete analysis of the potential links between dust in snow and snow instability, highlighting two situations: an impact on a dry mid-winter snowpack and an impact on the onset of wet-snow avalanches in spring. On the one hand, a potential effect of dust deposition on dry-snow slab avalanches strongly depends on the timing of the dust deposition. For instance, if the dust layer is immediately buried by a thick layer of clean snow, its radiative impact will be minimal in the days following the deposition and its impact on avalanche danger as well. Conversely, if the dust layer remains at the surface under clear sky conditions, the albedo decrease can induce surface melting which would have not happened without dust deposition. In the latter case, if the melted surface contaminated with mineral dust is buried under a thin layer of cold snow, a strong temperature gradient can form between the warm dust layer and the cold snow surface. This situation is similar to documented situations where persistent weak layers adjacent to crusts promote low stability (Jamieson, 2006; Colbeck and Jamieson, 2001; Birkeland et al., 1998). In the Pyrenees, for instance, a persistent weak layer located above a dust layer caused many avalanches in 2014 although in that particular case the dust layer and the instability may well be unrelated (Chomette et al., 2016). Both Landry (2014) and Chomette et al. (2016) reported that this short-term impact is not systematic and strongly depends on the timing of the deposition and on the subsequent weather conditions. On the other hand, a dust layer at the snow surface can increase the solar energy uptake by the snowpack (e.g. Painter et al., 2012) and induce stronger melting rates. As a consequence, wet-snow instabilities may form earlier in the season (Landry, 2014; Toepfer et al., 2006). Moreover, dust layers in the snowpack can re-appear at the surface when the overlaying snow layers melt (Doherty et al., 2013; Zhao et al., 2014). Mineral dust at the surface has an even stronger radiative impact in spring at the end of the snow season and speeds up ablation leading to a shorter snow season (Landry, 2014).

Despite these interesting processes observed in the field, it is not possible to isolate with certitude the impact of dust. It is challenging to demonstrate how snow instability is linked to the dust deposition only using field observations since it would require a "reference" snowpack without dust (Chomette et al., 2016). In order to investigate the significance of the physical processes described above, we use here a numerical modelling framework to assess whether an impact on snowpack stability can be attributed to strong dust deposition events. To this end, we use the recent developments in the ensemble version of the detailed multi-layer snow cover model Crocus-MEPRA (Brun et al., 1989; Vionnet et al., 2012) which allows us to represent the interactions between light absorbing particles and snow metamorphism (Charrois et al., 2016; Tuzet et al., 2017) and enables us to calculate snow instability indicators (presented in Section 2). This modelling set-up makes it possible to run the same simulation (topographic and meteorological conditions) with and without dust deposition in different topographic configurations to assess

the impact of dust on snowpack stability. The results of the numerical experiments are then presented in Section 3 and discussed in Section 4. The aim of the numerical experiments is to study the influence of the dust deposition on snow instability rather than to study the associated meteorological conditions.

## 2 Methods

### 2.1 Ensemble snowpack modelling framework

The modelling chain SAFRAN-SURFEX/Crocus-MEPRA Morin et al. (2020) provides the meteorological conditions for a given mountain region. The meteorological data are then used to simulate the snowpack in the mountain regions and to eventually assess the mechanical stability.

First, the meteorological forcing is produced by the SAFRAN meteorological analysis system. SAFRAN computes the weather conditions at hourly intervals across the French mountain ranges by analyzing meteorological surface observations from various networks Vernay et al. (2022). In the presented simulations, two types of light absorbing particles are considered: dust and black carbon. Black carbon deposition fluxes are forced by the regional climate model ALADIN-Climate (Nabat et al., 2014; Drugé, 2019) and dust deposition fluxes are adjusted as explained in Section 2.2.

Second, snow cover simulations are performed with the detailed snow cover model Crocus which simulates snow physical properties by computing the mass and energy exchange within the snowpack and between the snowpack, the soil and the atmosphere (Vionnet et al., 2012). Recent developments to represent light absorbing particles in Crocus Tuzet et al. (2017) facilitate computing their radiative impact with the TARTES (Two-stream analytical radiative transfer in snow; (Libois et al., 2013)) radiative transfer model. Note that the activation of this option is the main difference with the operational set-up described in Morin et al. (2020).

Uncertainties in snowpack models are related either to the atmospheric forcing or to the representation of snow physical processes (Krinner et al., 2018; Raleigh et al., 2015; Essery, 2013). In the present study, it is essential to account for the errors in the snow physical processes since the targeted processes are expected to be sensitive to other snow physical processes (Landry, 2014). In order to address this issue, a multi-physics ensemble modelling framework called ESCROC (Ensemble System CROCus) was developed for Crocus (Lafaysse et al., 2017). ESCROC is an ensemble of parameterizations of the snow cover model providing estimates of the uncertainty due to the representation of the main simulated physical processes. Each set of parameterizations is called a member. Note that the uncertainties in the meteorological forcing are not accounted for.

Finally, MEPRA, a simulation support tool for avalanche forecasting that is in operational use in France, is used to assess the mechanical stability of simulated snow profiles (Giraud, 1992).

### 2.2 Simulation set up

#### 2.2.1 Synthetic dust event

Two ensemble snowpack simulations of ESCROC are performed with an output time step of 3 hours. The set of parameterizations of both ensembles are identical, except for the forcing of mineral dust deposition. The first ensemble, called the no-dust simulation or no-dust ensemble, is run without dust deposition for the entire season. The second ensemble simulates a synthetic single dry dust deposition event on the 5 March 2018. We chose this date as we expected the meteorological conditions to allow for a potential impact of the dust deposition on the snowpack properties. It also brings the advantage that all the targeted processes can be studied on a single case. Dust deposition was simulated with a constant deposition flux leaving $8.6\,\mathrm{g.m^{-2}}$ of dust at the surface. The value was chosen for the synthetic event as it is representative of a strong but realistic Saharan dust outbreak in the region of interest (Réveillet et al., 2021). For both ensembles, the black carbon deposition fluxes were obtained from the ALADIN-Climate model. For the dust ensemble simulation, TARTES computes the spectral albedo considering both black carbon and dust, while in the no-dust ensemble the evolution of the albedo is calculated considering black carbon the only light absorbing particle present in the snowpack. The comparison between the dust ensemble and the no-dust ensemble provides a numerical estimation of the impact of the dust deposition event on the evolution of snow properties as well as the associated modelling uncertainties. The ESCROC ensemble used here, called E2_Tartes, has already been used in Tuzet et al. (2020); Dumont et al. (2020) and is an adaptation of the ensemble E2 which is fully described in Lafaysse et al. (2017). This ensemble is composed of 35 members and the TARTES radiative transfer scheme is used for all the members of our study as it is the only one to explicitly account for the impact of light absorbing particles. Therefore, the modelling uncertainties on the radiative transfer scheme are not accounted for in this study. We also neglect the uncertainty in black carbon deposition and the impact of dust deposition for the rest of the season.

Simulations are carried out in the Thabor mountain region at 2400 m elevation. This region in the French Alps is close to the Italian border (region number 13 in fig. 2 of Vernay et al., 2022). All presented simulations were conducted on a slope inclined by 40°. Eight different aspects are computed: north, northeast, east, southeast, south, southwest, west or northwest, as in the operational forecasting system.

To investigate the sensitivity of the results to elevation and to the intensity of the dust deposition, additional ensemble simulations with several elevations and varying mass of deposited dust. The following deposition masses: $8.6\cdot10^{-2}$ , $8.6\cdot10^{-1}$ , $4.3\cdot10^{+1}$ $\mathrm{g.m^{-2}}$ correspond to typical values of Saharan dust deposition in the French Alpine regions ranging from a low Saharan dust outbreak to an extreme case with a deposition mass 50% higher than maximum measured values from Réveillet et al. (2021). The snow-modelling uncertainty is still considered in this sensitivity experiments with the same ensemble framework as described in Section 2.1.

#### 2.2.2 Observed dust event

The method presented above was also applied to an observed major dust outbreak event that occurred from 5 to $7^{th}$ February 2021 (Réveillet et al., 2021). Over the Thabor massif, where our synthetic case is performed, loads of dust ranging from 6.16 $g.m^{-2}$ to 29.7 $g.m^{-2}$ were measured on 12 snow samples with a median value of 14.9 $g.m^{-2}$ (values from data used in Réveillet et al. (2021)).

The simulations were performed following the same methodology as for the synthetic case (see above) with a deposition of 15 $g.m^{-2}$ to be consistent with measured concentrations. The black carbon deposition flux was set to the median value of ALADIN-Climate on year 2017-2018 because ALADIN-Climate output was not available for this period.

### 2.3 Impact evaluation

To assess the impact of the simulated dust outbreak on snowpack stability, different snow physical properties and stability indicators are compared between the no-dust and dust simulations. We study dry-snow instabilities in view of artificial triggering and wet-snow instabilities for natural release, as they are subject to dust deposition (Landry, 2014).

#### 2.3.1 Artificial triggering of dry slab avalanches

The impact of the dust deposition on the probability of artificial triggering in the region is evaluated using the MEPRA stability indicator. The model MEPRA analyses at every time step the output of the snow cover model, calculates the snow mechanical properties of the snow layers (e.g. shear strength or ram resistance), and based on that data, provides stability indicators for the probability of natural release or artificial triggering. Figure 11 of Morin et al. (2020) shows an example of a final product of the analysis with MEPRA as it is provided to French avalanche forecasters.

The so-called "accidental risk index" in MEPRA provides an estimate of the probability of artificial triggering due to an additional load at the snow surface (e.g. due to a recreationist) on a four level scale. On this scale, the level 0 indicates that the initial conditions for instability, i.e. a typical weak layer underneath a cohesive slab, aren't fulfilled. Level 3, the highest level, on the other hand, corresponds to pronounced instability where weak layer strength is close to the stresses in this layer.

To calculate the "accidental risk index" the MEPRA model searches for a slab (a layer of decomposing and fragmented precipitation particles or rounded grains), sitting on top of a weak layer, i.e. a layer of faceted crystals (FC), depth hoar (DH), precipitation particles (PP) or decomposing and fragmented precipitation particles (DF). Then, MEPRA compares the shear strength to the shear stress in the weak layer (due to the weight of the overlying slab and skier induced stress) (Giraud et al., 2002; Viallon-Galinier et al., 2022). The stress in the weak layer is adapted to include so-called bridging effects, for instance, the effect of a melt-freeze crust that redistributes the

stresses due to surface weights (Giraud et al., 2002; Thumlert and Jamieson, 2014). For operational forecasting the "accidental risk index" is combined with a "natural risk index", we removed this tie and only consider the part described above. Hence, in dry-snow situations the described MEPRA "accidental risk index" can be considered an indicator of the probability of artificial triggering. In the following we refer to it as the MEPRA index.

The MEPRA stability indicator, $M$, is computed for each output time step of the simulation, $t$, i.e. 3 hours, and for every member $i$ of both, the dust and no-dust ensemble. For both ensembles the total number of members is set to $N = 35$ as in Lafaysse et al. (2017) so that the spread of the ensemble adequately represents the model uncertainties. In order to compare both ensembles, the stability indicator of the dust simulation, $M_{\text{dust}}$, is subtracted from the one of the no-dust simulation, $M_{\text{no_dust}}$, member by member. For each member, $i$, and time step, $t$,

$$\Delta_{M,i}(t) = M_{\text{dust},i}(t) - M_{\text{no_dust,i}}(t), \quad (1)$$

is positive, if the stability is higher in the case of the dust simulation than in the case of the no-dust simulation. Values of $\Delta_{M,i}(t)$ range from -2 to 2. $\Delta_{M,i}(t)$ is computed from the first time step of the dust deposition until 10 April 2018 at 6 a.m. in order to ensure the presence of snow in both simulations for each index $i$ and $t$ considered here. This period corresponds to 312 simulation time steps, i.e. 312 values of $t$.

We define the probability $P(V,\Omega_t)$ (expressed in %) as the probability that the stability indicator $\Delta_M = V$ takes a certain value $V$ in a given time domain $\Omega_t$ among the ensemble members. For instance, for a given time step, $t$, if 7 of the members present a $\Delta_M(t)$ of 1, $P(1,t) = 20\%$ (7 values over a total number of 35).

In Section 3, we use daily and seasonal values of $P(V,\Omega_t)$. The daily values are computed for each day from midnight to 9 p.m. as a mean on temporal time steps and members. For the seasonal values, we only use the sign of $\Delta_M$, and calculate for each member we the number of days when $\Delta_M$ was positive or negative. We report the distribution within the ensemble of more stable and less stable days with respect to the no-dust simulation.

Finally, in order to quantify the bridging effect, we also use the bridging index (Thumlert and Jamieson, 2014) for relevant weak layers: for all slab layers, the product of RAM penetration resistance (computed by Crocus, in daN (Giraud et al., 2002, Section III.1)) and layer thickness (in cm) is computed, and values are summed up to provide one value for the slab.

#### 2.3.2 Onset of the first wet-snow avalanche cycle

The impact of dust deposition on wet snow avalanche activity is evaluated using variables simulated with Crocus. In this study we use the liquid water content index ($\text{LWC}_{\text{index}}$) as introduced by Mitterer et al. (2013). The $\text{LWC}_{\text{index}}$ is calculated by dividing the mean volumetric liquid water content of the simulated snowpack by a typical value of $0.03\,\text{kg.m}^{-3}$. In order to identify the onset of wet avalanche activity, this index is compared to 0.33 according to Mitterer et al. (2016): the first wet-snow avalanches can be expected, when the liquid water content index reaches this value.

## 3 Results

### 3.1 Meteorological conditions

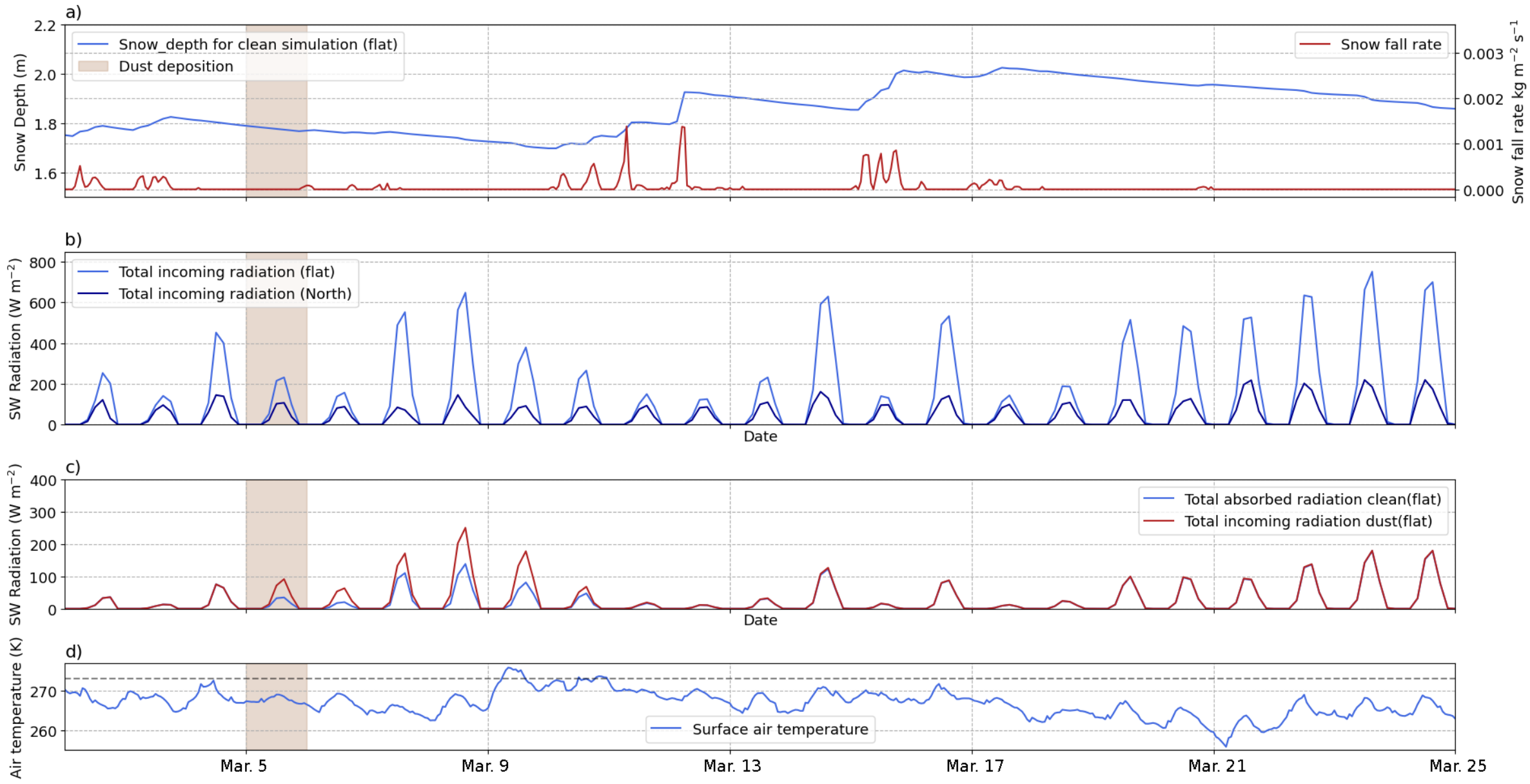

**Figure 1.** Meteorological and snowpack conditions in the days following the dust deposition on 05 March 2018: a) Snow depth for the no-dust simulation (flat field) and snowfall rate; b) total amount of incoming solar radiation for a flat field and a 40° steep north-facing slope d) air temperature. Panel c) presents the energy absorbed by the snowpack for dust and no-dust simulations (member 8) on a flat surface.

On the 5 March 2018 conditions were favorable for dust depositions to have an impact on snow instability according to Landry (2014), i.e. the dust deposition was followed by a dry period and enough incoming shortwave radiation. Figure 1 presents the meteorological conditions in the days following the dust deposition. In this case, dust was deposited at the snow surface and stayed at the surface for five days. During these five days the shortwave incoming radiation on a flat surface was moderate on two days (6 and 10 March) and high with mostly clear-skies from 7 to 9 March (Figure 1b). The dust radiative forcing (i.e. the additional energy absorption due to the surface darkening of dust) was up to 100 W.m$^{-2}$ in peak hours (Figure 1c). The dust layer was then buried by new snow of around 30 cm of new snow that fell on 11 and 12 March 2018. The radiative impact of the dust layer became negligible during the following two weeks (Figure 1c) when a cooler period followed which is reflected in negative values of air temperature.

### 3.2 Dry-snow instabilities

#### 3.2.1 Impact on snow profiles

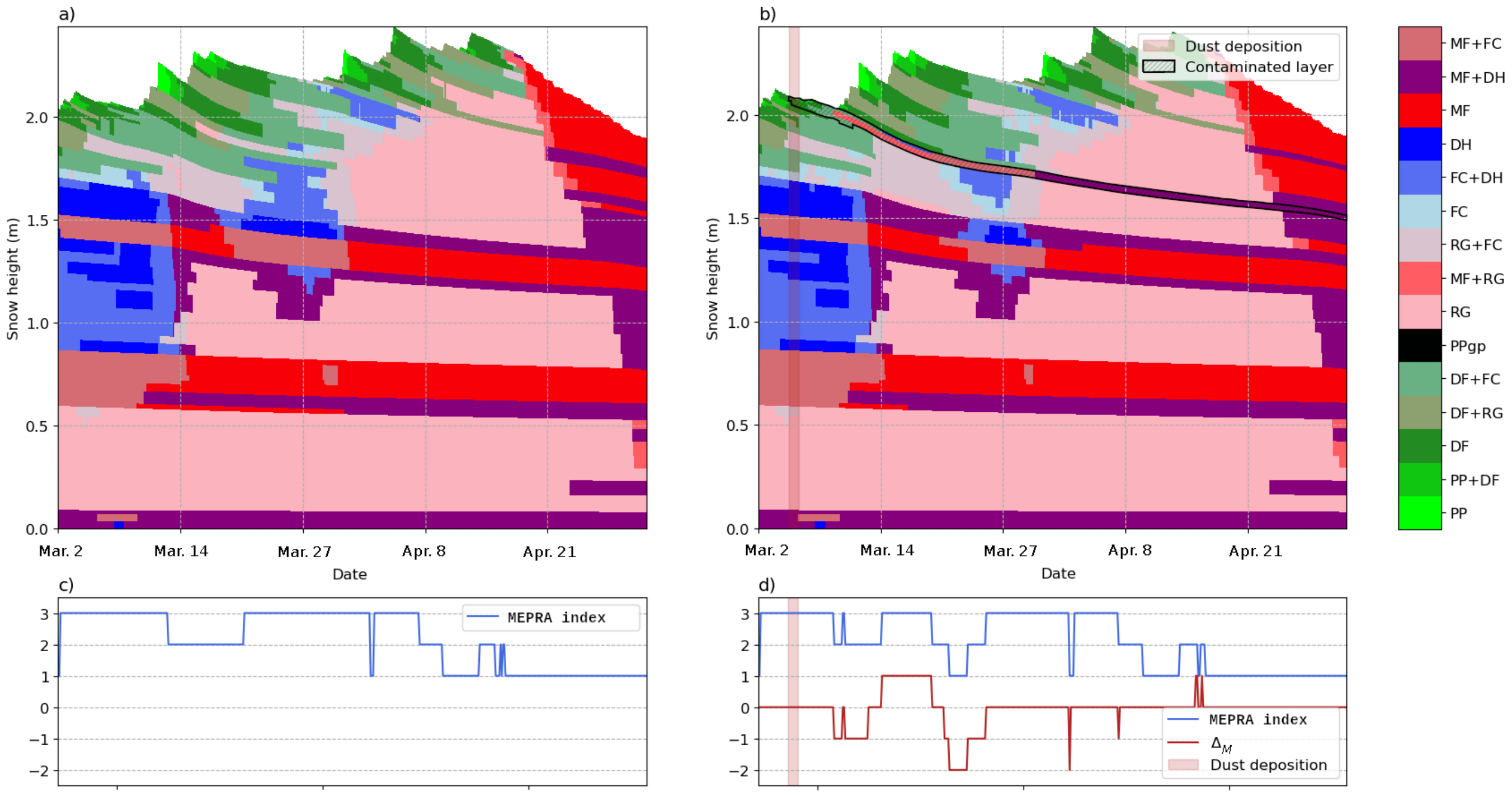

**Figure 2.** The upper panels show snow profiles for the same member (member 8) of the ensemble simulations on a 40° steep north-facing slope. (a) refer to the no-dust simulation and (b) to the dust simulation. Snow grain shape abbreviations follow the international snow grain shape classification (Fierz et al., 2009) (Table A1), the shading (hatches) corresponds to the layer contaminated with more than $10^{-4}$ g.g$^{-1}$ of dust. The lower panels present the MEPRA stability indicator (in blue) for no-dust (c) and dust (d) simulations. Panel d) also shows in red the $\Delta_M$ index which represents the difference of stability between both simulations.

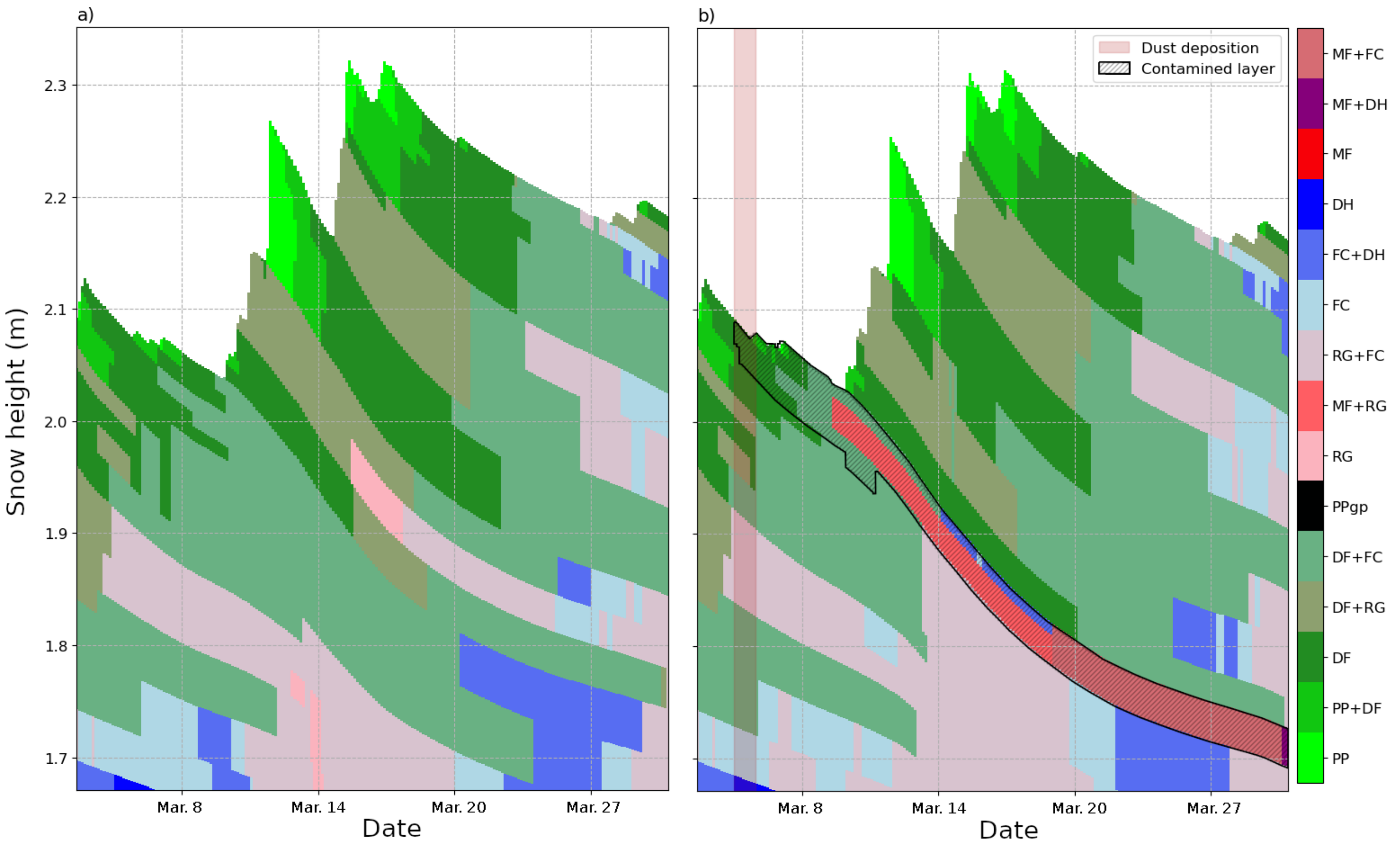

**Figure 3.** Zoom of Figure 2

Figures 2 a) and b) show the temporal evolution of of grain shape profiles for member 8 of both ensemble simulations for a 40° steep north-facing slope, respectively with and without dust. The snow cover in the dust simulation (Figure 2b) has a twenty-five millimeter thick melt-freeze crust which appears four days after the dust deposition just below the surface (see also Figure 3 for a zoom on the interesting period). This melt-freeze crust does not exist in the snow cover simulation without dust deposition (Figure 2a) and hence, was caused by the additional surface melting induced by the dust layer which was at the snow surface between 5 and 9 March (Figure 1). The cold weather conditions after the snowfall induced a temperature gradient in the upper part of the snowpack and conditions were favorable for faceting adjacent to the crust. Four days after the snowfall, on 14 March, a thin layer of faceted crystals and depth hoar had formed above the melt-freeze crust in the dust simulation. This can be explained by a strong temperature gradient ($>20\,\mathrm{K.m^{-1}}$) between the dust layer and the overlying recent snow (not shown), which was not present in the no-dust simulation.

Figure 2 c) and d) present the evolution of the MEPRA stability indicator. Figure 2 d) also shows the evolution of $\Delta_M$, and presents the difference of the stability indicators between dust and no-dust simulation. In the four days following the snowfall, both scenarios show poor stability due to the presence of faceted crystal and depth hoar

around 30 cm below the surface. Just before the snowfall, the formation of the melt-freeze layer reduced the stress due to a potential skier on the underlying weak layer via a bridging effect, also reducing the probability of artificial triggering in the dust simulation for four days. On 24 March, the bridging index for the weak layer which is just below the crust in the dust simulation is 114 daN.cm for the simulation without dust and 162 daN.cm with the melt freeze crust created in the simulation with dust, which represents an increase of 42% of the bridging effect, according to this index. Then, the appearance of a new weak layer of faceted crystals (FC+DH) decreased stability in the dust simulation. As this layer did not form in the no-dust simulation, the dust simulation was less stable than the no-dust simulation for five days before it was merged with adjacent layers, as snow structural properties allowed the model to do so. In the days following 19 March the probability of artificial triggering decreased in the dust simulation and became lower than in the no-dust simulation. This may be due to the crust which redistributes stresses inside the snowpack so that the weak layer stress is lower. After 25 March the impact of dust deposition became negligible as snow instability was controlled by more recent temporal weak layers.

Figure 4 shows the snow profiles on 18 March for all 35 members, in order to represent the sensitivity of the stratigraphy to snow modelling uncertainties. The member 8 shown in Figure 2 is highlighted by a black rectangle. Not all the members exhibit the same behaviour as member 8. For some members, the net surface energy budget does not provide the energy to melt the dust layer located around 1.8 m above the ground (hatched rectangle in figure 2b). Even in this case, the temperature increase caused by dust can be sufficient to cause the growing of a weak layer of faceted crystals (e.g. Figure 3). For the dust simulation, the formation of a melt-freeze crust can be observed for 30% of the members, while it never occurs in the no-dust simulation. More than 48% of the members present a faceted layer (FC, FC+DH or DH) around 1.8 m above the ground in the dust simulation, compared to 23% in the no-dust simulation.

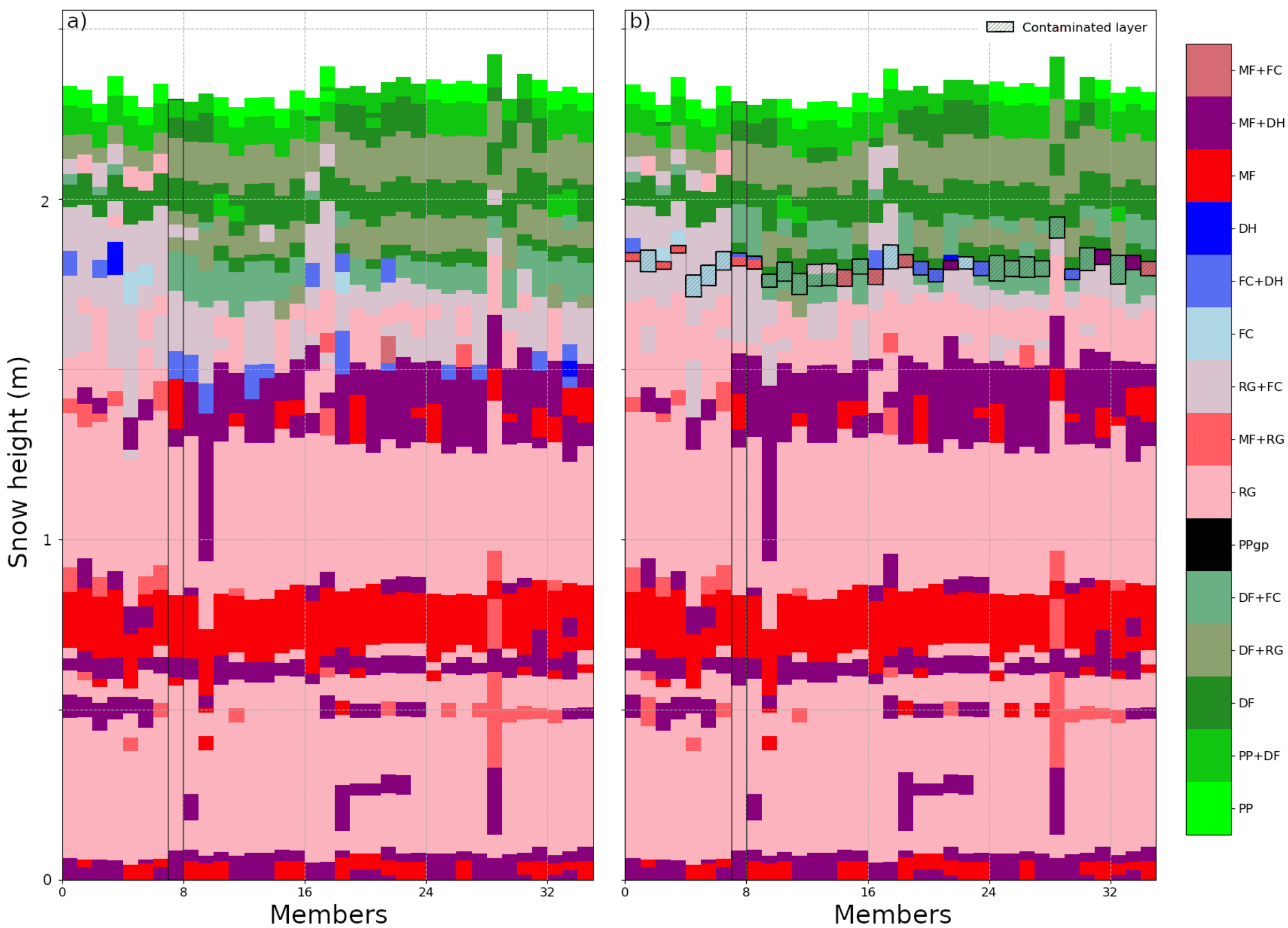

**Figure 4.** snow profiles of the 35 members of the no-dust simulation (panel a) and dust simulation (panel b) for the northern slope on the 18 March, at noon. The member represented in Figure 2 is surrounded by a black rectangle and the contaminated layers for each member are hatched in panel b. Snow shape names follow the international snow grain shape classification (Fierz et al., 2009).

### 3.2.2 Ensemble stability analysis

Figure 5 presents the daily bar plot of P(V,day) (the probability that $\Delta_M = V$ each day amongst the 35 members (Section 2.3.1) for north-facing aspects. The probability of no impact (P(0,day)) is not represented but corresponds to the complement to 100%. A positive value $\Delta_M$ means that the computed stability is lower for the dust simulation than for the no-dust simulation and a negative value means that the stability is higher for the dust simulation.

From 9 to 11 March, more than 20% of the snowpack simulations in the ensemble have a negative $\Delta_M$. This can be attributed to the presence of a melt-freeze crust in the dust simulation that decreases the stress on the underlying weak layers as explained in section 3.2.1. Afterwards, on 12 March more than 20% of the snowpack simulations have a positive $\Delta_M$ for 7 consecutive days. This corresponds to the period of enhanced temperature gradient and grain

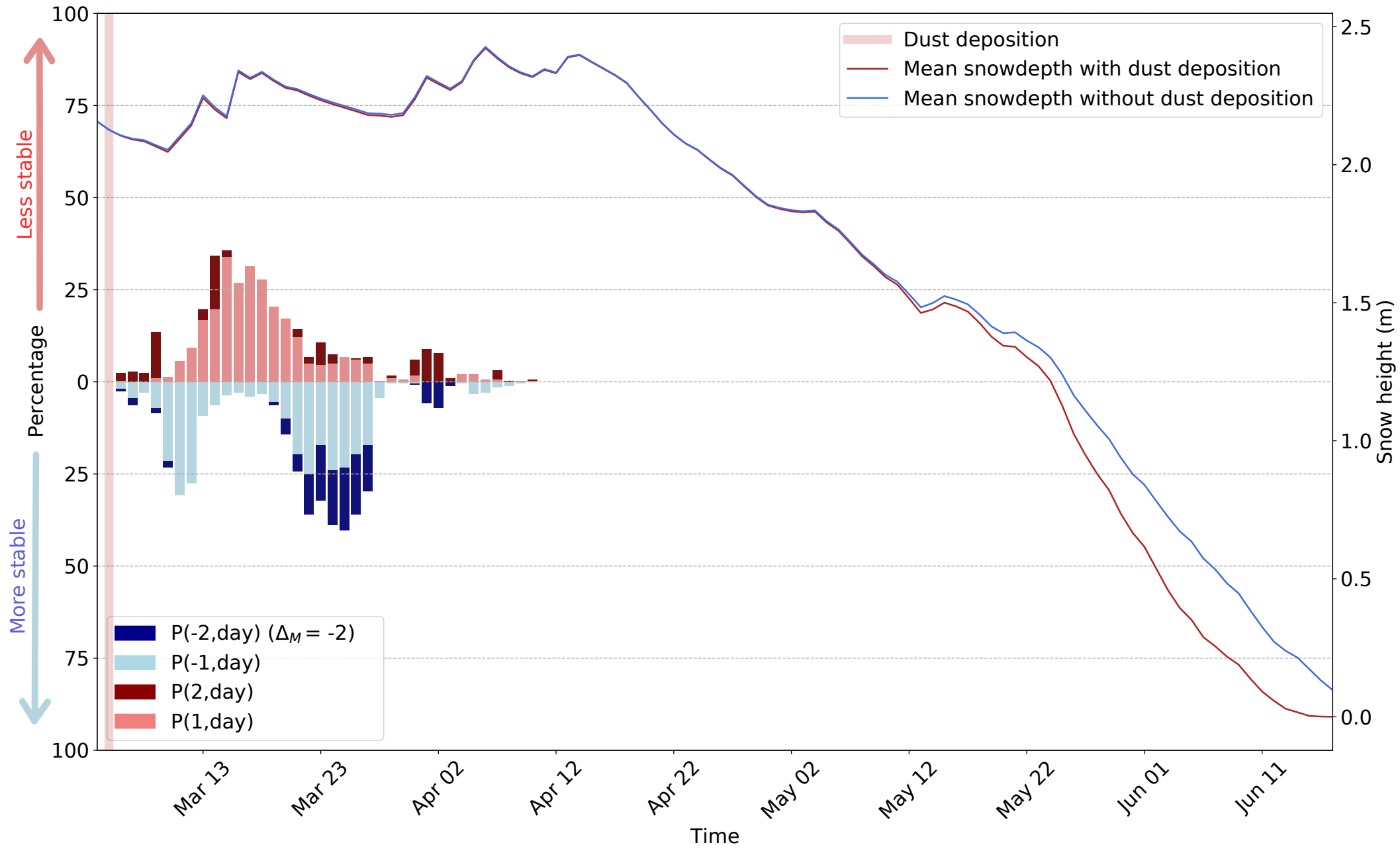

**Figure 5.** Evolution of the daily difference of stability $\Delta_M$ for the synthetic case of a north-facing slope at 2400 m. For each day the probability $P(V,(day))$ (in %) is graphed in blue for an increase in snowpack stability ($V < 0$) and in red for a decrease ($V > 0$). The blue and red lines represent the snow depth for the modelled snowpack without and with dust deposition, respectively.

faceting identified in Section 3.2.1. On 20 March a period of eight days begins when more than 25% of the snowpack simulations were more stable in the dust simulations. After 27 March, the dust and no-dust simulations show no major difference in stability. Snow height values of both simulations show small differences a few days after the dust deposition and no systematic difference can be highlighted until May 11th. After May 11th, the snow cover with dust deposition melts faster. Figure 5 highlights that the impact of dust deposition alternates between an increase and a decrease in snow stability in the weeks that follow dust deposition. Although a consistent temporal signal can be identified among the ensemble members, the complex interactions between processes add a high level of uncertainty to the simulation of the impact of dust deposition, as at any time 60% of the members exhibit no change in MEPRA indicators.

### 3.2.3 Impact of the deposited mass of dust

To investigate the sensitivity of the results of subsections 3.2.1 and 3.2.2 to the dust deposited mass, several ensemble simulations were performed with different dust masses. Four different masses are tested: $8.6 \cdot 10^{-2}$ , $8.6 \cdot 10^{-1}$ , 8.6

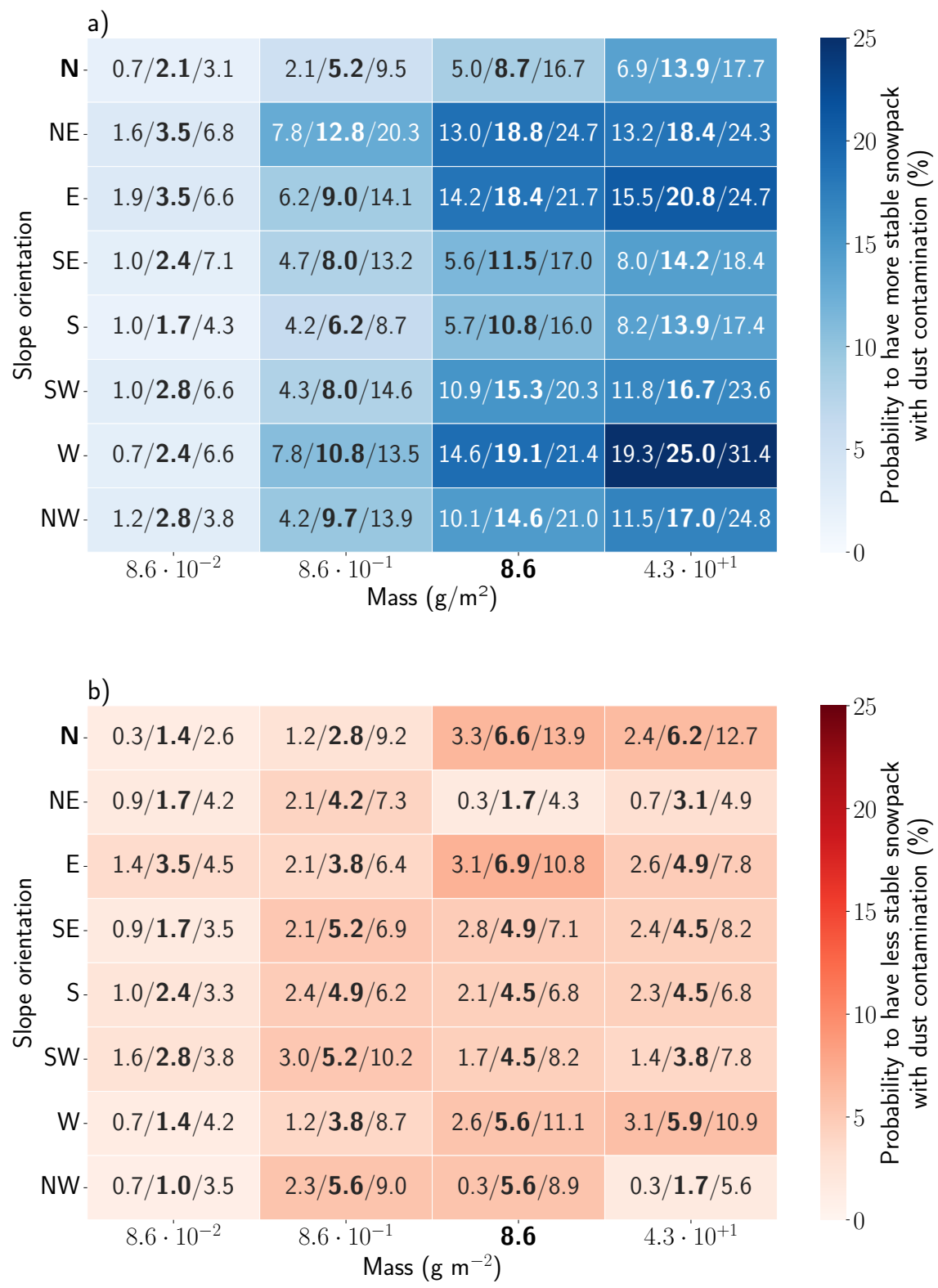

**Figure 6.** Probability for a more stable (negative $\Delta_M$) (a) and a less stable (positive $\Delta_M$) (b) snowpack due to dust deposition on the eight slope aspects for four different dust deposition masses: $8.6 \cdot 10^{-2}$ , $8.6 \cdot 10^{-1}$, 8.6 , $4.3 \cdot 10^{+1}$ g.m$^{-2}$. The $\Delta_M$ index represents the difference of dry-snow stability between dust and no dust simulations. The values are calculated from 5 March to 9 April (280 timesteps) corresponding to P(<0,all) for (a) and P(>0,all) for (b). The deposition value of 8.6 g.m$^{-2}$ correspond to the previously studied configuration (Sections 3.2.1 and 3.2.2). The values indicated in the cells are the median (in bold) and the first and third quartile. All values are percentages referring to the probability for a negative or a positive value.

(corresponding to the initial study case presented in previous results) , $4.3 \cdot 10^{+1}$ g.m$^{-2}$. The probability to have more stable (negative $\Delta_M$; Figure 6a) or less stable (positive $\Delta_M$; Figure 6b) snowpack in the dust simulation than in the no-dust simulation is computed using the method described in section 2.3.1 from 5 March (date of dust deposition) to 9 April.

Figure 6a) shows that for all aspects increasing the amount of the deposited dust tends to increase the probability for lower dry-snow stability in the dust simulation. This is explained by the higher dust radiative forcing leading to more melting and a more pronounced bridging effect of the melt-freeze layer (Figure 2). Regarding the less stable days, Figure 6 b) illustrates a complex link between the dust mass deposited and the topographic conditions with no evident relationship between these two factors and the number of less stable days. This is consistent with previous observations highlighting the strong variability of this impact (e.g. Landry, 2014). This shows that the impact of dust towards lower dry-snow stability is not negligible under certain conditions (deposition mass, topography, other terms of the surface energy budget...). In other words, our data do not suggest a rule to explain how stability depends on conditions.

#### 3.2.4 Impact of elevation and aspect

The influence of elevation and slope aspect on the results of Section 3.2.1 and 3.2.2, was estimated by calculating for each ensemble member the number of days when the snowpack is more stable (Figure 6a) or less (Figure 6b) stable in the dust simulation than in the no-dust simulation. Eight slope aspects at three different elevations 2100 m, 2400 m and 2700 m were considered.

Figure 7a) shows no marked trend even if the number of more stable days seems to be lower at 2100 m than at higher elevation. This can be attributed to the presence of a melt-freeze crust in all snowpack simulations (dust and no dust) at this elevation. Figure 7b) presents the variability of the impact of slope aspect and elevation. No clear trend was apparent, but the impact varied strongly between two neighboring configurations (e.g. N at 2400 m and NE at 2400 m). The dispersion between the 25th and 75th percentiles (small plots in Figure 7) also shows that the number of days when the stability was impacted by dust can vary significantly with the accuracy of the simulated energy balance, as estimated by the ensemble modelling framework, from a few days to more than 30 days at a given elevation elevation and aspect.

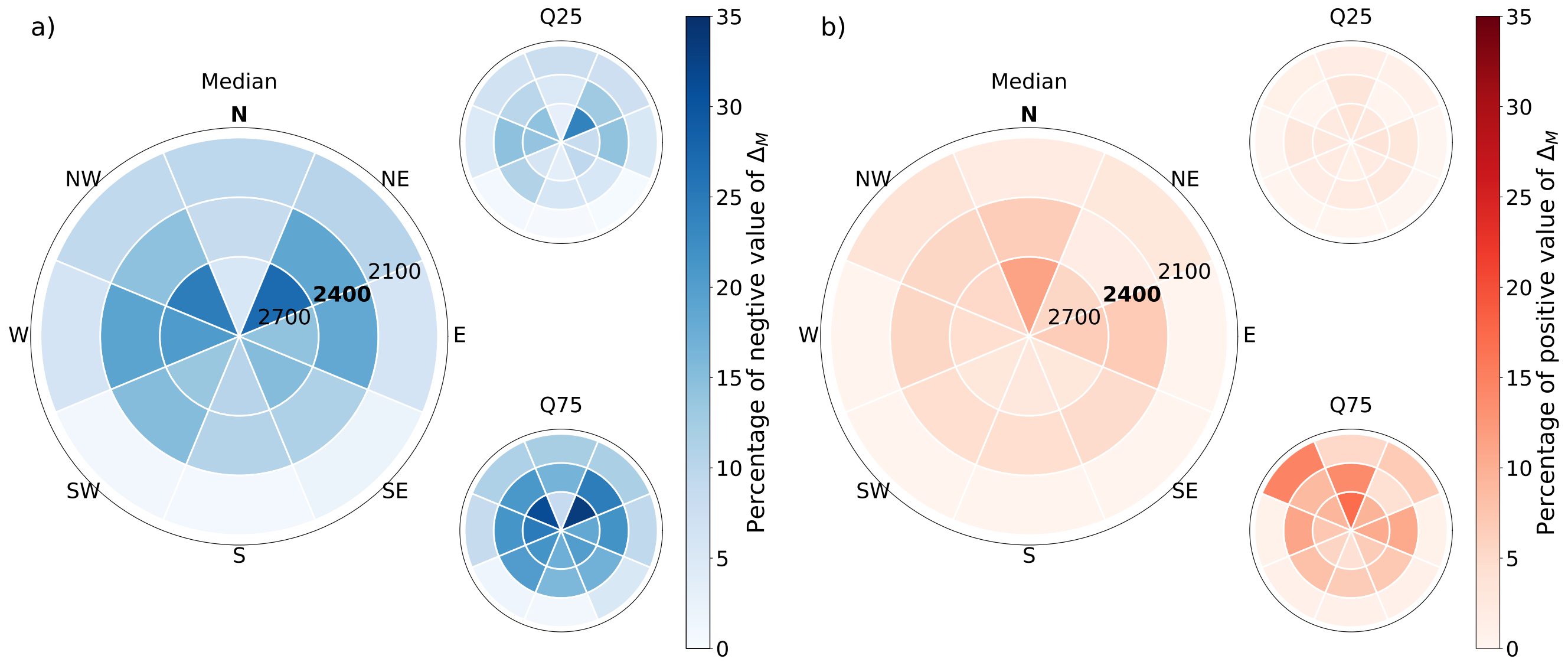

**Figure 7.** Number of days with (a) a more stable and (b) a less stable snowpack when contaminated with dust, for each of the eight slope aspects and for three different elevations: 2100 m, 2400 m, 2700 m. The values are calculated from 5 March to 9 April. The bold labels (2400 m north) correspond to the configuration of Sections 3.2.1 and 3.2.2. The biggest pie plot corresponds to the median of ensemble members and the small pie plots to the first (Q25) and third (Q75) quartile of ensemble members.

### 3.3 Wet-snow instabilities

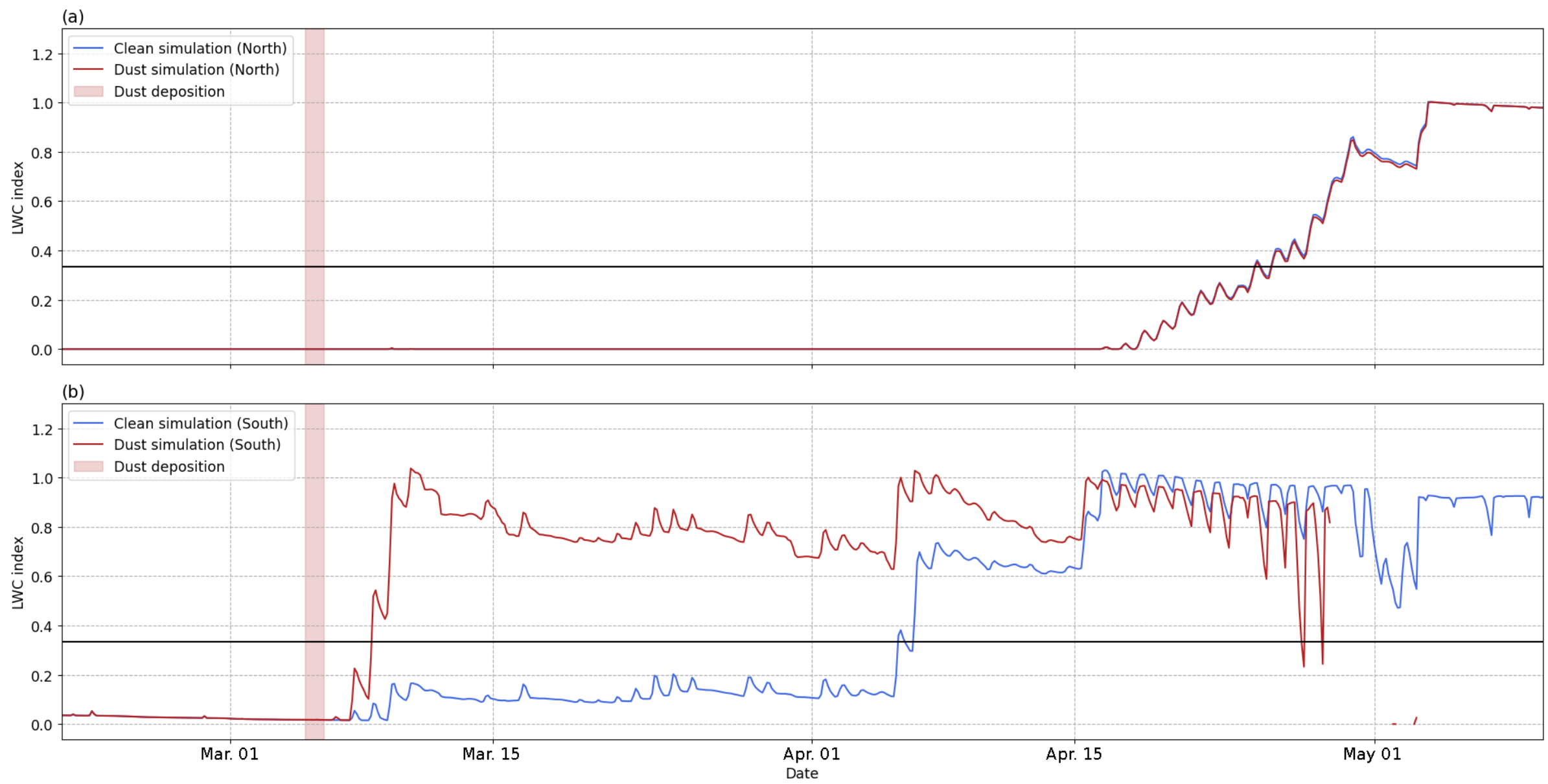

**Figure 8.** LWC index for member 8 of both dust and no-dust simulations for on a 40° steep north-facing (a) and south-facing slope (b). The horizontal black line shows the threshold of 0.33.

Figure 8 illustrates the evolution of the LWC index for season 2017-2018, as described in Section 2.3, for the member 8 of both dust and no-dust simulations. The horizontal black line at 0.33 represents the lower threshold used to assess the onset date of the first wet-snow avalanches (Mitterer et al., 2016).

On the north-facing aspect (Figure 8a), the LWC index reached 0.33 on 24 April for both no-dust and dust simulations. Both simulations were close meaning that the impact of the prescribed dust deposition on the wet-snow avalanches onset is negligible for this aspect. This can be attributed to the weak incident solar radiation on slopes with such an aspect, which seems insufficient to cause melting in the snowpack even in presence of dust. Some small differences appear later in the season between both simulations (not shown), but there is still no impact on the data at which the threshold is exceeded. In contrast, for the south-facing aspect (Figure 8b) the threshold value of LWC is reached on 8 March for the dust simulation and 27 days later, on 4 April, for the no-dust simulation. The trend of both members matches again on 16 April. This can be interpreted as the onset of wet-snow avalanches advanced by 27 days due to dust deposition for the considered member.

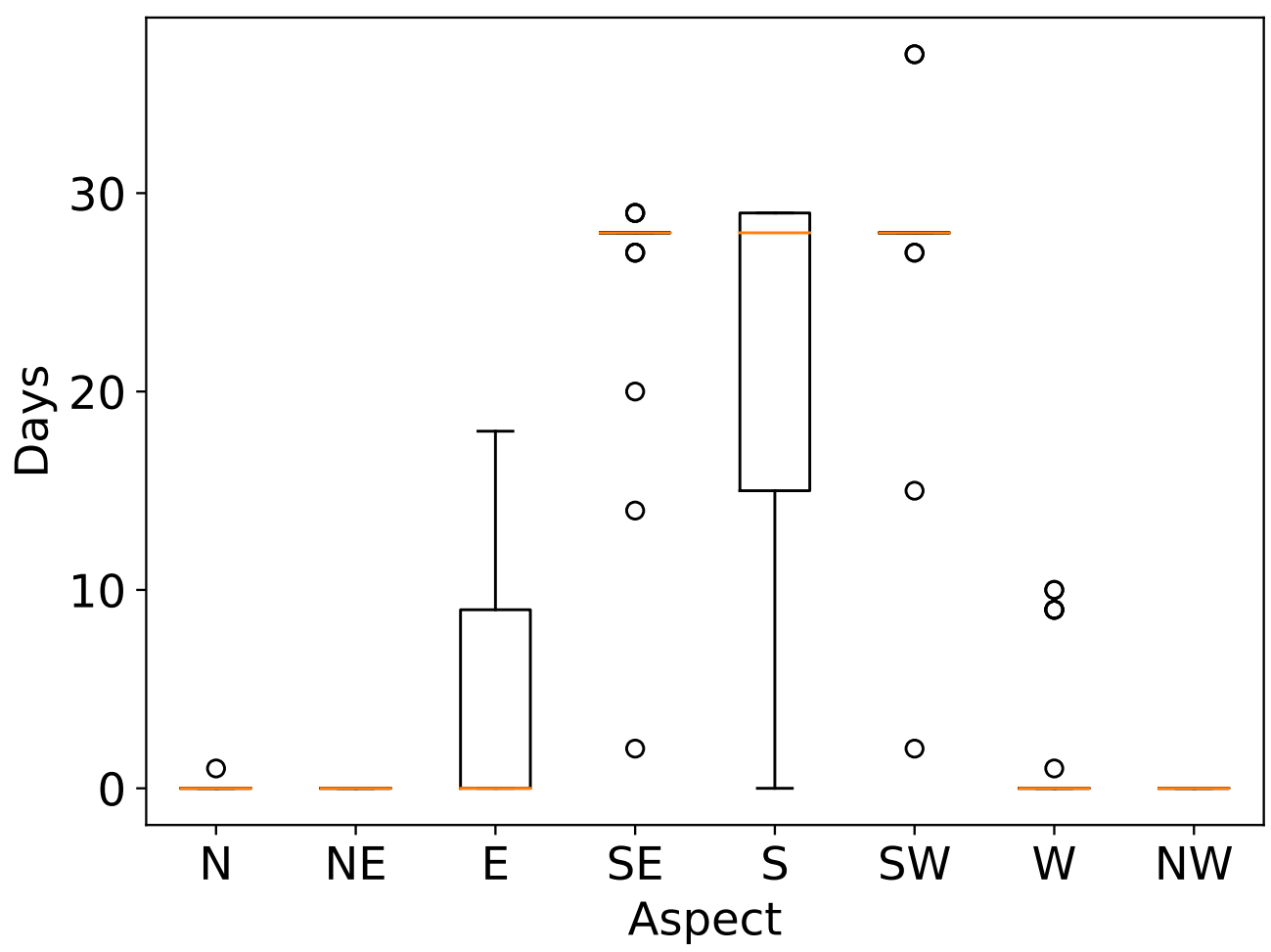

**Figure 9.** Impact of the dust deposition on reaching the LWC threshold. For each aspect, the difference in date when the threshold was reached was calculated between the dust and the no-dust simulation. The main body of the boxplot spans the interquartile range from the first to the third quartile of the data, while the horizontal orange lines show the median. The whiskers show the range of observed values that fall within 1.5 times the interquartile range and the black dots represents outliers.

When extending the analysis on the whole ensemble, we also observe the impact of dust on the onset date of wet-snow avalanches on south-facing slopes. Figure 9 features the difference of date between the time that the LWC exceeds the threshold in the no-dust and the dust simulations. For the south-facing slope, 25% of the snowpack simulations have a difference of less than 15 days while about 50% have a difference of 28 days or more with a median value of 29 days. This means that the dust deposition leads to a shift of almost one month of the onset date of wet-snow avalanches for south-facing aspects. The members featuring a difference of less than 5 days correspond to members for which the threshold was already almost exceeded at the date of deposition (Figure A1). This situation is less common on southwest and southeast-facing slopes, as the incoming shortwave radiation is lower. For these aspects, the spread is extremely limited with more than 94% of the members showing an advance of the onset of wet snow avalanche of more than 27 days. On east-facing slopes, more than 50% of the snowpack simulations show no difference, while 25% have a difference of 9 days or more. For west-facing aspects, most of the members show no impact, while 20% of the members have a difference of 9 days or more. Finally north, northeast and northwest-facing slope simulations show no impact. For these aspects, results show that dust has no impact on when the threshold is reached. This is likely related to the lower relative impact of shortwave radiation on the total surface energy budget, meaning that snow melting on shady aspects is mainly induced by other terms of the surface energy budget.

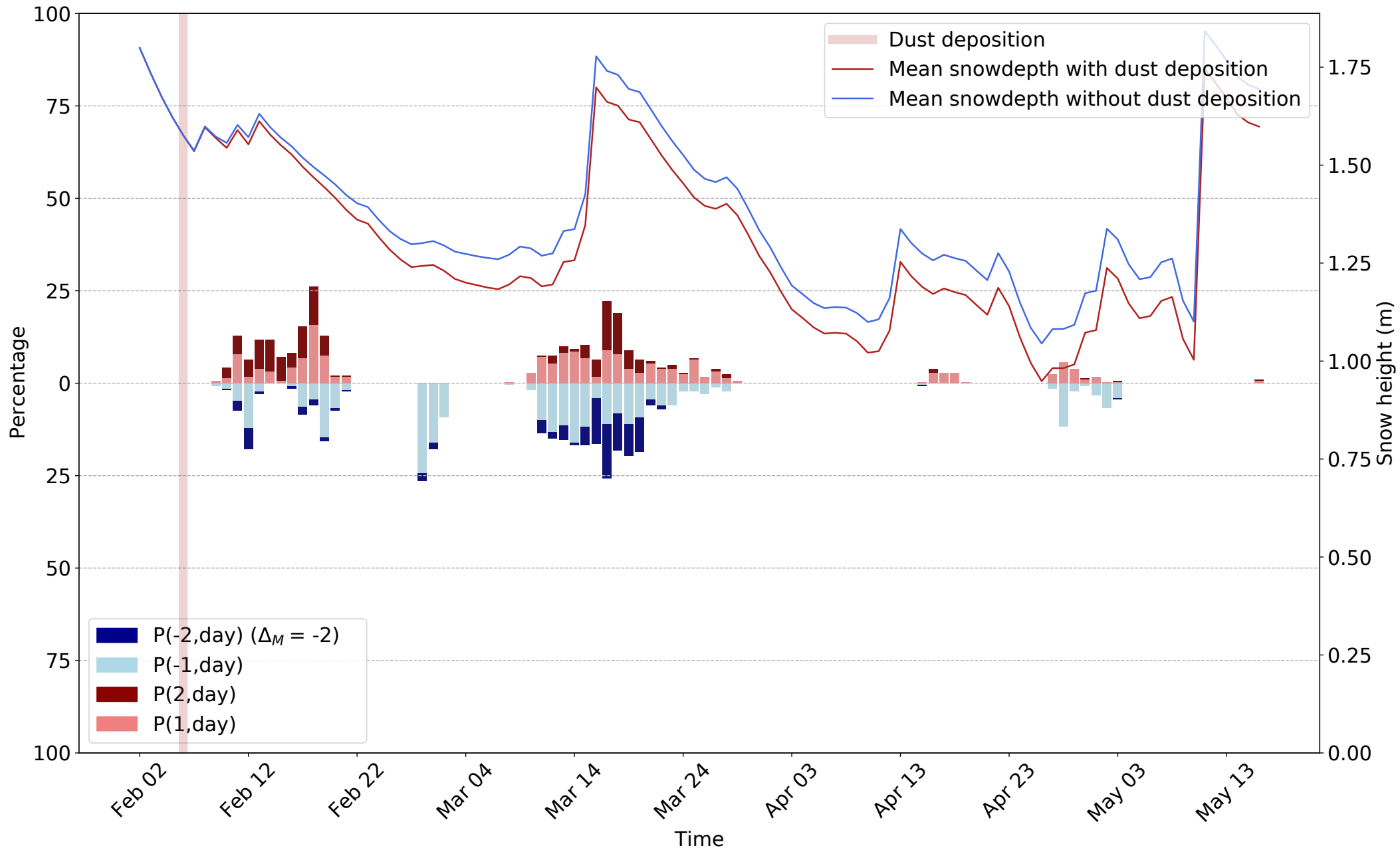

**Figure 10.** Evolution of the daily $\Delta_M$ for a north-oriented slope at 3000 m and for the observed dust event of February 2021.

### 3.4 Dust event of February 2021

Figure 10 presents the results on a west-facing slope at 3000 m highlighting the dust deposition event of February 2021. It shows that the dust deposition event mainly modified dry-snow stability for two periods, right after the deposition event and after the consequent precipitation event mid-March, slightly more than one month after the deposition.

Figure 11 displays the number of days with more or less stable slope for this deposition event. The methodology is the same as for Figure 7. It shows that the impact is close to zero for south-facing slopes but is more pronounced for north and east-facing slopes, at all elevations. However, the magnitude of the impact (either more stable or less stable) is lower than for the synthetic case (Figure 7).

## 4 Discussion

The present study focuses on the impact of dust outbreaks on snowpack stability, a subject that has been debated by practitioners for a long time with no clear scientific answer to date. The subject has already been treated qualitatively and Landry (2014) highlighted two main typical situations: an impact on dry-snow instability and an impact on wet-

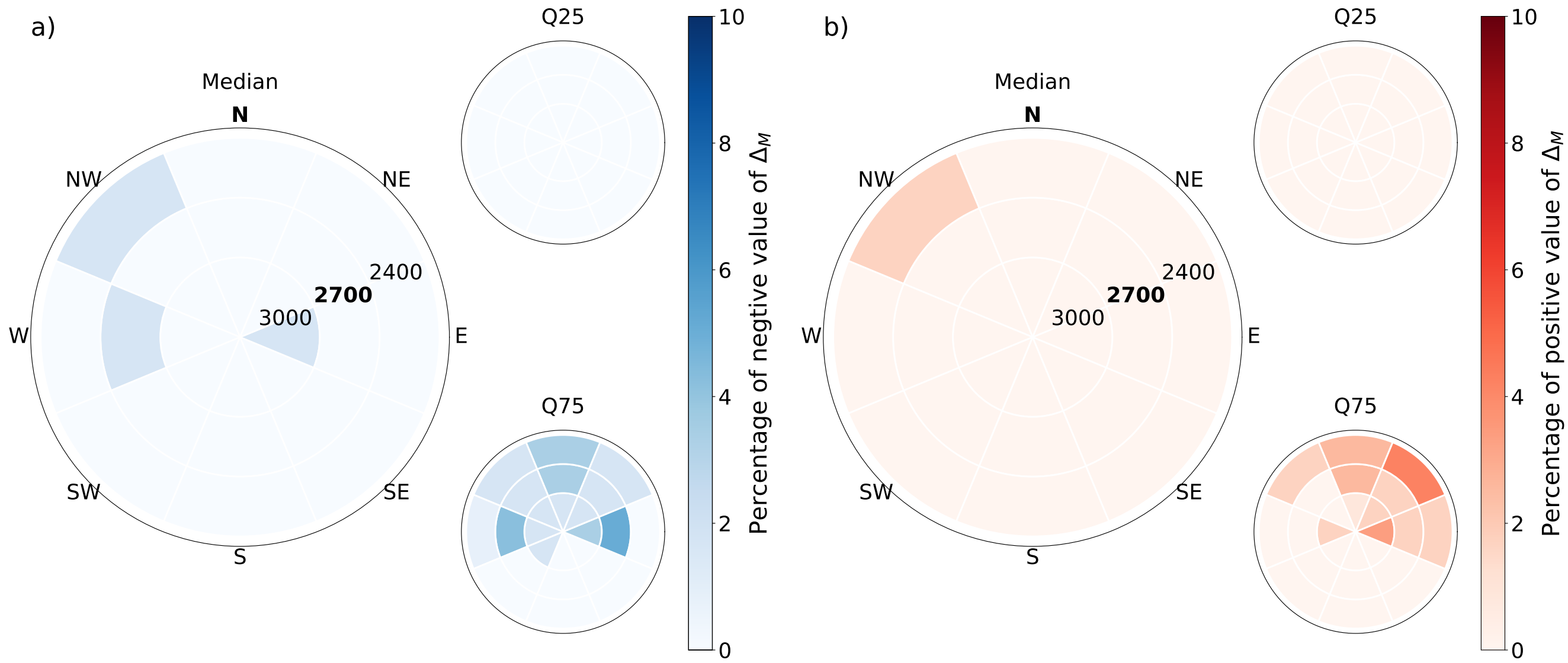

**Figure 11.** Number of days with (a) a more stable and (b) a less stable snowpack when contaminated with dust, for each of the eight slope aspects and for three different elevations: 2100 m, 2400 m, 2700 m. The analysis period covers from the dust event to the 6 April. The $\Delta_M$ index is introduced in Section 2.3.1 and represents the difference of dry-snow stability between dust and no dust simulations. The biggest pie plot corresponds to the median of ensemble members and the small pie plots to the first (Q25) and third (Q75) quartile of ensemble members.

snow instability. To our knowledge, no clear evidence of these processes have been demonstrated until now, partly due to the difficulty to set up an experimental case (Chomette et al., 2016). To address this issue, we use a multi-physical snowpack modelling approach with the ESCROC snowpack model to separate the impact of dust from the impact of other meteorological factors by comparing simulations with and without dust deposition. A synthetic case with a dust deposition in the Thabor massif on 5 March 2018 (dust simulation) was compared to a similar ensemble simulation without any dust deposition (no-dust simulation), making it possible to separate the impact of dust from associated meteorological conditions. The two processes described by Landry (2014) were reproduced numerically. First, we show that the dust deposition induces an impact on dry-snow instability with alternating phases of increase and decrease of snowpack stability. This impact and its sign has a strong dependency on the deposited dust mass and the topographic conditions (Subsection 3.2.2 as hypothesized by Landry (2014) and Chomette et al. (2016)). Second, an advance of the onset date of the first wet-snow cycles in spring was observed. The shift is larger for higher incoming shortwave radiation (Section 3.3). The ensemble simulations were also performed for the major dust deposition event observed in February 2021 (Réveillet et al., 2021). The simulations show that, in some cases, the

dust deposition can have an impact on the simulated stability indicators (either to more stable or to more unstable) even one month after the deposition event. However, the magnitude of the impact, in terms of number of days when stability was affected, is at least 3 times lower than for the synthetic case.

### 4.1 Dust impact on dry-snow avalanche formation

The modelling experiment presented in sections 3.2.1 and 3.2.2 shows that dust can significantly affect the surface energy budget to alter snow metamorphism. For some members, the increase of solar absorption induces the formation of a melt-freeze crust that would not have formed in absence of dust (Figure 2). In the studied case, we point out processes that can both increase or decrease snowpack stability. The increase of snowpack stability is relatively intuitive and may relate to the formation of a melt-freeze crust contaminated by dust in the snowpack that reduces the stress on an underlying weak layer by bridging effect (Schweizer et al., 2003). The effect of bridging effect was quantified with the help of the bridging index from Thumlert and Jamieson (2014). The decrease of snowpack stability comes from an enhanced temperature gradient adjacent to the crust as our synthetic case showed. The impact observed appears to be particularly sensitive to factors such as slope aspect, elevation, meteorological conditions and deposited dust mass (Section 3.2.2) both for the synthetic and for the observed cases. Depending on the aforementioned parameters, a given dust deposition can either lead to an increase or a decrease of snowpack stability or have no impact on the dry-snow avalanche danger. The simulated impact of dust on stability is also highly dependent on the uncertainties in the simulated energy balance coming from the other processes represented in the model, as shown by the member-dependent impact in Section 3.2.2 and the large ensemble spread in Fig. 11. All these results are consistent with the observations of Landry (2014) and Chomette et al. (2016) that both underlined that the impact is not systematic. This suggests that the predictability of the sign, magnitude, and localization of this impact is particularly challenging.

For several members of our ensemble simulation, the additional energy absorption caused by dust leads to the apparition of a melt-freeze crust at (or near) the surface of the snowpack (e.g. Figure 2). The formation of sun crusts (refrozen wet layers initially formed by absorbed solar radiation) for clean snow is discussed in Section 2.3 of Jamieson (2006). The formation of sun crusts have been reported to be highly sensitive to topographic variables (slope, aspect and elevation) and to be difficult to predict by avalanche forecasters (Jamieson, 2006). We show here that the phenomenon can be amplified by the presence of dust due to the increase of solar radiation absorption. The dependency of the dust impact on topographic variables (results of Section 3.2.2) is expected to add complexity to avalanche forecasting Landry (2014). For instance, when solar energy input is too weak to induce the melt of a clean snowpack, conditions may still be favorable for melting if light absorbing particles are present.

The presence of a weak layer of faceted crystals above a melt-freeze crust has already been documented and can be considered as a typical situation for slab avalanche release (Jamieson et al., 2001; Jamieson, 2006). The formation of such a weak layer is due to the following physical processes, the strong temperature gradient between the crust and the overlaying snow favours the kinetic growth of faceted crystals which is further enhanced due to the low thermal

conductivity of the faceted layer in relation to the melt-freeze layer (Colbeck and Jamieson, 2001; Hammonds et al., 2015). This last point can be simulated by Crocus snowpack model, which can reproduce temperature gradients around crusts to form weak layers. However, the gradients at the millimeter-scale, that seem to play a role in the weakening of faceted layers adjacent to crusts (Hammonds et al., 2015; Hammonds and Baker, 2016) cannot be reproduced by the model's vertical resolution. Hence, the weak bonding between the melt-freeze crust and the faceted layer which can be conducive to slab avalanche release (Jamieson, 2006) may likely be underestimated. This means that the decrease of snowpack stability due to the dust could be more pronounced than the one our simulations predict when a melt-freeze crust is forming.

According to the simulations obtained in the synthetic and observed dust deposition events, the impact of dust on snow instability is not limited to the period following the deposition. When dust stays exposed at the surface and helps forming a crust, as in the February 2021 case, the impact of the buried crust can be detected weeks later (Figure 10).

### 4.2 Dust impact on wet avalanches

Regarding the wet-snow avalanche activity the results are systematic. When the dust layer re-appear at the surface at the end of the season, the induced reduction of albedo causes an earlier wet-snow avalanche activity in the season. As expected, this advance increases with the incoming shortwave radiation which explains the strong impact on south-facing aspects, can be as pronounced as 30 days (Figure 8, Figure 9). These findings confirm that the dust has an impact on the surface albedo and solar radiation absorption inducing an earlier onset of the wet snow avalanche season. These results agree with observations of Landry (2014) in the Rocky Mountains of Colorado. On the other hand, the north-facing slopes did not receive enough solar energy to impact the timing of wet-snow avalanches in our synthetic case. The impact in north-facing slopes is also expected to vary according to the timing of the usual wet-snow season in absence of dust. Indeed, in our study case, the simulation on North slope exceeds the liquid water content threshold around 25 April and at this period, 40° north-facing slope does not receive enough solar radiation to significantly impact the timing of melt. In these aspects, other terms of the surface energy balance such as latent heat release or the longwave radiation drive snow surface warming (Reuter and Schweizer, 2012).

### 4.3 Limitations

Our work provides numerical evidences that dust deposition can modify snowpack stability in the French Alps. However, some limitations related to our approach remain.

We used the MEPRA stability indicator for estimation of the dry-snow stability and implemented the $LWC_{index}$ for wet snow, while a large variety of other stability indicators exists (Viallon-Galinier et al., 2022). The MEPRA indicator used here is based on discrete values which can limit the accuracy of the comparison between two simulations. For instance, there are many days in our analysis for which the stability indicator is maximal for both the dust and the no dust snowpacks due to the presence of deep weak layers in both cases (e.g. Figure2 d). In such

case, the potential impact of dust on surface stability is missed and the $\Delta_M$ is null. For example, the approach of Reuter et al. (2022) can help to circumvent this limitation by tracking the weak layers over time and assessing the avalanche problem types based on their stability. Moreover, the wet-snow instability indicator used in our study is sensitive to the liquid water percolation scheme used in the model. The discrepancies between members are especially marked here for east- and south-facing aspects (Figure 9), and can be partly explained by the three different options implemented in the ensemble to model the maximum liquid water retention capacity of snow (Lafaysse et al., 2017). Ensemble modelling makes it possible to assess the sensitivity of our results to the liquid water percolation scheme highlighting a clear impact on slopes of a large South sector. However, an improvement of the representation of this complex three-dimensional process in snow models might reduce the associated uncertainty and improve the characterization of wet snow stability (Wever et al., 2018).

Another limitation lies in the fact that our study is restricted to two cases: the observed case of a major dust deposition event in February 2021 and a synthetic dust deposition case selected since it provides a good illustration of both negative and positive impact on snowpack stability. Beyond both cases presented here, dust depositions have been tested for other years at several dates (not shown) highlighting a high sensitivity of the impact to the date, aspect, dust load, elevation, and snow model uncertainty. This high sensitivity confirm the results already described for the two cases investigated in details in this study.

In addition, we only consider the impact of dust on the snow optical properties and its consequences on energy exchanges between the snowpack and the atmosphere, whereas dust could possibly have other impacts on the snow cover. For instance, Meinander et al. (2014), Skiles and Painter (2016) and Seidel et al. (2016) provided some observational evidence of non-radiative impact of light absorbing particles in wet snow, namely changes in liquid water retention capacity and metamorphism which also deserve to be further investigated as they could be important processes in this specific question.

Finally, the results obtained here translate to other types of light absorbing particles, deposited in high enough concentration to have similar radiative impacts. In French mountain ranges, dust exhibits the strongest sporadic deposition but in other regions, the dust outbreaks presented here could be compared in a way to volcanic eruption that deposit large amounts of ashes at once over the snow surface.

## 5 Concluding remarks

This study is a first approach to investigate the impact of dust on snowpack stability by numerical modelling. The modelling approach makes it possible to separate the impact of dust from the impact of other meteorological variables, which would be challenging in a field experiment. We numerically investigated the impact of dust outbreaks on both, dry-snow and wet-snow instabilities. The impact of dust was studied in a synthetic and an observed dust deposition case. Snow modelling uncertainties were considered with an ensemble snow cover modelling framework.

Regarding wet-snow instability, using the liquid water content index proposed by Mitterer et al. (2016), we confirm that dust causes an earlier onset of the first wet avalanche cycles in the snow season for slopes with a sufficient dust-induced surface melting (south-facing slopes). In our study case, the predicted onset of the wet-snow avalanche season advanced by up to 1 month due to the presence of dust. These results agree with the observations of Landry (2014) in the Rocky Mountains.

Concerning the probability of artificial triggering in dry-snow, we identify three possible scenarios due to dust layers: no impact, a decrease of snowpack stability, meaning that the dust renders the snowpack less stable compared to the no-dust simulation, and an increase of snowpack stability due to the deposited dust. In some meteorological conditions, for instance when the dust layer is not exposed to solar radiation and is directly buried by fresh snow, the presence of dust can have no impact on snowpack stability.

In our synthetic case, the dust layer stayed at the surface of the snowpack for 5 days before being buried by 30cm of fresh snow. When dust is at the surface, it reduces snow albedo, which enhances surface warming, which can cause the formation of a melt-freeze crust while in the absence of dust a crust hadn't formed. Once covered with new snow, a strong temperature gradient can form around the crust, resulting in the formation of faceted crystals or depth hoar. In our simulations some members exhibited this weak layer formation process, that decreased snowpack stability. On the contrary, the crust that formed due to dust-enhanced surface warming can increase snowpack stability by reducing the stress in the weak layer as it helps redistributing forces laterally and making failure initiation less likely.

Whether the balance tips towards increasing or decreasing snowpack stability depends on the intensity of dust-induced surface melting, and therefore strongly depends on the deposited dust mass, the slope aspect, the elevation and the weather conditions following the dust outbreak. To conclude, there is no simple answer to the question: "Would this avalanche have occurred without the dust outbreak?". The main conclusion is that dust deposition can indeed impact snowpack stability, even though several meteorological and snow cover conditions need to line up to promote instability. Our simulation for the observed case suggests that only in a few cases dust deposition will decrease snowpack stability.

An analysis of dust events extended to a longer period, possibly in different snow climates, can shed light on the likelihood of such events. The available snow cover models can reproduce the influence of dust deposition on snow stratigraphy and snowpack stability. However in future snow cover models, the faceting process adjacent to crusts could be refined to pinpoint changes of snow stability.

*Data availability.* The meteorological data used to drive snow simulations are available at Vernay et al. (2020). Processed data are available upon request to the corresponding author.

*Code and data availability.* The Crocus model is open-source and the code is available at https://opensource.umr-cnrm.fr/projects/snowtools_git/wiki/Procedure_for_new_users. The version used is labelled as s2m_reanalysis_2020.2. The configuration of the ensemble of models used is described in details in Lafaysse et al. (2017).

*Author contributions.* Oscal Dick, Léo Viallon-Galinier, François Tuzet and Marie Dumont designed and wrote the study. Oscar Dick, Léo Viallon-Galinier, François Tuzet and Mathieu Fructus ran the simulations. Matthieu Lafaysse and Pascal Hagenmuller participated to the data analysis and Benjamin Reuter provided the data analysis with an alternative stability indicator. All authors contributed to the writing and commenting of the manuscript.

*Competing interests.* M. Dumont is Editor for The Cryosphere. The authors declare that they have no other conflict of interest.

*Acknowledgements.* CNRM/CEN is part of Labex OSUG@2020. Marie Dumont and François Tuzet have received funding from the European Research Council (ERC) under the European Union's Horizon 2020 research and innovation program (IVORI (grant no. 949516)). This work was partly funding by ANR JCJ EBONI (ANR-16-CE01-0006) and APR CNES MIOSOTIS.

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

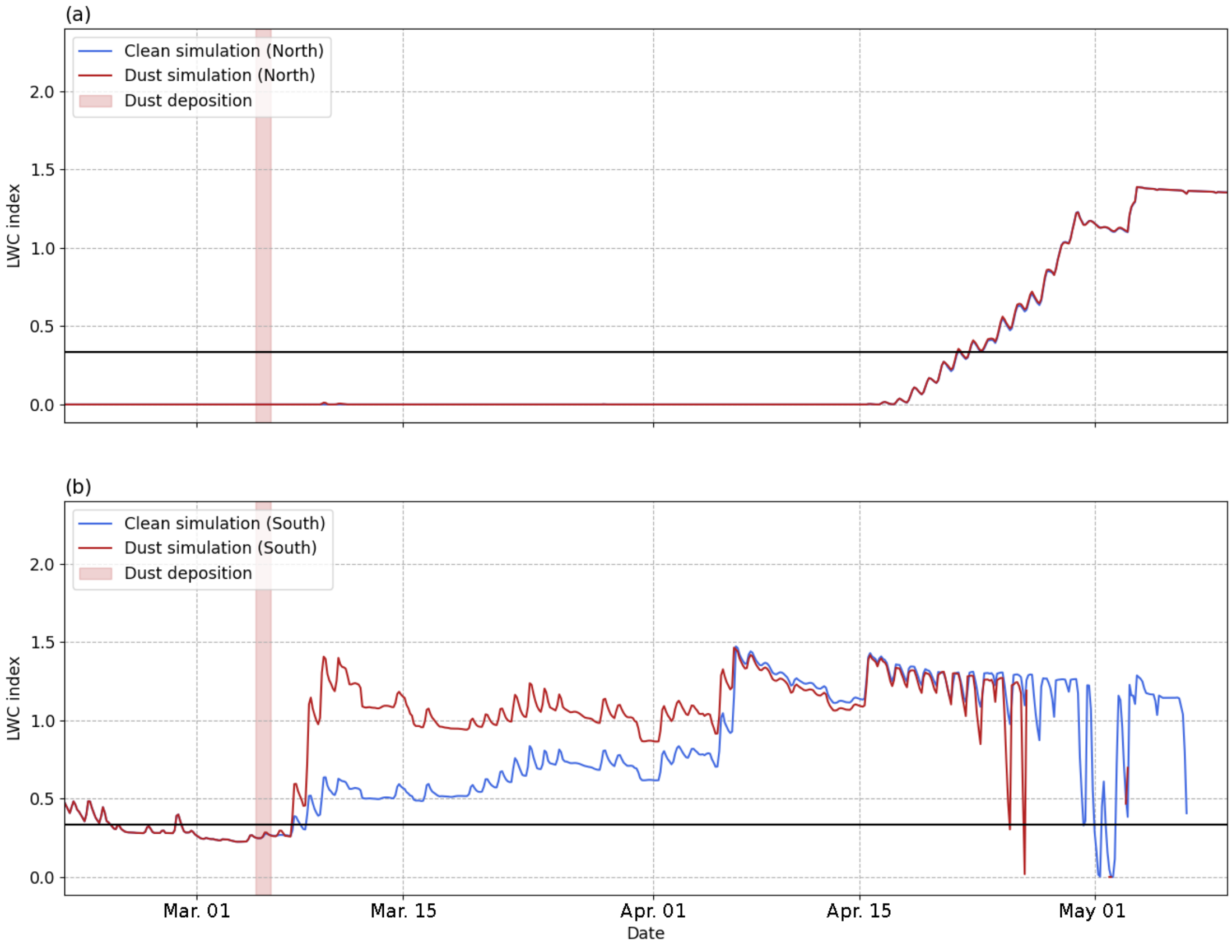

**Figure A1.** Same representation as Figure 8 for another member of the ensemble (member 20) featuring a LWC index already high at the deposition timing

| Code | Name of the class |
|---|---|
| PP | Precipitation particles |
| PPgp | Graupel |
| DF | Decomposing and fragmented precipitation particles |
| RG | Round Grains |
| FC | Faceted crystals |
| DH | Depth Hoar |
| MF | Melt forms |

**Table A1.** Table of the standardized grain shape classes used in this study. See Fierz et al. (2009) for detailed description of each class.