# Peer review of "Can Saharan dust deposition impact snowpack stability in the French Alps?"

_The Cryosphere, 2022_

## Referee Comment (RC1)

To the authors and TC,

I provide here a review of the article "Can Saharan dust deposition impact snowpack stability in the French Alps?", submitted to The Cryosphere (TC-2022-219). My recommendation is for publication in TC pending only minor revisions.

**Summary**: In this study, an ensemble snow cover model was used to investigate the impact of dust deposition on snow properties and mechanical stability by comparing simulations with and without dust deposition for one synthetic and one observed dust deposition event. The modelling chain, used to perform the simulations, employed SAFRAN-SURFEX/Crocus-MEPRA for meteorological conditions, ALADIN-Climate for dust deposition fluxes, and TARTES for radiative transfer calculations in snow. Although each of these models bear with them their own set of uncertainties, not addressed explicitly within the study, the multi-physics ensemble modelling framework ESROC (Ensemble System CROCus), an extension of the snowpack model CROCUS, was used to account for uncertainties in the snow physical processes expected to drive the results of the simulations, given all other model inputs/outputs were held constant between the two comparative cases of interest (clean vs. dust-contaminated snow). To evaluate mechanical stability, a stability indicator was derived from MEPRA, a simulation tool that is used operationally in France to assess the mechanical stability of simulated snow profiles. After establishing the framework, the authors used this model-chain to thoroughly evaluate both dry snow and wet snow conditions over a variety of aspects and elevations for their numerical study site, the Thabor Massif, which was also applied to one case-study of an actual dust-on-snow event that occurred in 2021.

**Strengths:** Although as a reviewer I am admittedly not an expert on all the inner-workings of the models mentioned above, I found the ensemble approach to handling snowpack property uncertainty and the use of a simple stability indicator from MEPRA convincing. Perhaps more importantly, I found the results of the study interesting and of broad significance to those attempting to better understand the role that light absorbing particles (LAP's) play in snowpack metamorphism for both climate-related and avalanche forecasting applications. For instance, the conclusion that LAP's deposited on the snow surface could act to both weaken and/or strengthen its mechanical stability of a dry snowpack, or that the spring wet snow avalanche cycle might occur in advanced timeframes on southern exposures, are both areas that are of relevance to operational avalanche or hydrological forecast programs. Given that there is currently a severe lack of controlled studies on this topic in the cryospheric sciences (numerical, laboratory, or field-based), to my knowledge, I applaud the authors in tackling this topic and providing a strong basis for future work in this area of research. Last, I particularly appreciated what seemed like an innovative approach to displaying the wide variety of statistical results produced, utilizing relatively easy-to-comprehend plots, elevation/aspect rosettes, and tables to present their data.

**Weaknesses**: The only weaknesses I found in the study, also acknowledged by the authors in the Discussion, was 1) that their study was limited to only one simulated and one actual dust-on-snow event, 2) that ESROC cannot resolve the vertical resolution of millimeter-scale snow processes that may occur at ice-snow interfaces, such as enhanced faceting, therefore rendering the MEPRA-derived stability indicator as somewhat limited in its regard for predicting some likely cases of snow instability, and 3) that weak layers were not tracked in such detail as to be

able to be continually reassessed as additional snowfall further buried them or compared against other persistent weak layers already in the snowpack prior to the dust-on-snow event (e.g. depth hoar).  This being said, given the scope of the study presented, and the abundance of data already needing to be parsed in their analysis, it seems acceptable to me that these potential weaknesses in the study only be acknowledged, which they were by the authors in the Discussion, and then perhaps left to be revisited in subsequent studies.

In conclusion, I found no major weaknesses in the study presented, and again applaud the authors on providing the snow research community with a detailed numerical study on a topic that is of both relevance and need to a broad set potential end-users.  Minor edits are given below.

Sincerely,

Kevin Hammonds
Director, Subzero Research Laboratory
Assistant Professor, Civil Engineering
Montana State University, Bozeman, MT, USA

**Minor Edits and Clarifications:**

Page 2, Line 5: I don't see a need to use the language, "The lack of a clear message…", when this is a very current area of research for multiple research groups in the cryospheric sciences.

Page 2, Line 15: recommend replacing "dropped off" with deposited.

Page 2, Line 26: "albedo decrease in the visible and near-infrared". Isn't snow highly absorptive in the NIR at 1500nm?  Such that dust impurities on the surface could actually increase the albedo?  Perhaps clarify via the wavelengths that are being referred to here and elsewhere in the article…or clarify that with your model you are only using the integrated spectral albedo and over which wavelengths.

Page 3, Line 2: Grammar…recommend deleting this sentence and extending the previous sentence with something like ", such that we only focus in the present study…"

Page 3, Line 3: Replace "Up to date," with To date.

Page 4, Lines 23,24, 27: Could you please expand upon what "snow physical processes" exactly are being evaluated and/or considered relevant to the remainder of the study?  As opposed to just referring to them somewhat generically?

Page 4, Line 28: Could you provide some examples of what meteorological forcing uncertainties are not accounted for? For instance, I'm curious about solar angle, diffuse vs. direct lighting conditions, localized heat advection in the Thabor Massif, etc.

Page 5, Line 8: Similar to above comment...what are the "targeted processes" exactly? Could you be more specific, where here or elsewhere these processes could be outlined more formally?

Page 5, Line 13,14: It is not clear to me why black carbon is used as an impurity in the no-dust ensemble, could you please elaborate? Is a constant background of some black carbon in the snowpack always assumed? What concentration? Why not run the clean-snow case without any impurities?

Page 5, Line 20: Do you have any ideas about modelling uncertainties on the radiative transfer scheme? Orders of magnitude or otherwise? Potential impacts of these and other uncertainties?

Page 6, Line 18: Could you please list all the snow mechanical properties that are output from MEPRA? As ooposed to "e.g., shear strength or ram resistance"…is it one or the other or both? More?

Page 7, Line 20: Delete ""we" from "member we the number"

Page 7, Line 30: Please provide a citation for 0.03 kg per m^3.

Page 10, Line 1: Delete "of" from "of of"

Page 22, Line 7: Delete "the one" from "than the one our"

Page 23, Line 1: Add "a case" to "case"

Page 23, Line 25: Add an s to "eruption", to make the plural "eruptions"

Page 23, Line 26: Delete "es" from "ashes".

---

## Author Comment (AC1)

**Answer to RC1 (Kevin Hammonds)**

Oscar Dick      Léo Viallon-Galinier      François Tuzet      Pascal Hagenmuller
Mathieu Fructus      Benjamin Reuter      Matthieu Lafaysse      Marie Dumont

**Main comments**

To the authors and TC,

I provide here a review of the article "Can Saharan dust deposition impact snowpack stability in the French Alps?", submitted to The Cryosphere (TC-2022-219). My recommendation is for publication in TC pending only minor revisions.

**Summary**: In this study, an ensemble snow cover model was used to investigate the impact of dust deposition on snow properties and mechanical stability by comparing simulations with and without dust deposition for one synthetic and one observed dust deposition event. The modelling chain, used to perform the simulations, employed SAFRAN-SURFEX/Crocus-MEPRA for meteorological conditions, ALADIN-Climate for dust deposition fluxes, and TARTES for radiative transfer calculations in snow. Although each of these models bear with them their own set of uncertainties, not addressed explicitly within the study, the multi-physics ensemble modelling framework ESROC (Ensemble System CROCus), an extension of the snowpack model CROCUS, was used to account for uncertainties in the snow physical processes expected to drive the results of the simulations, given all other model inputs/outputs were held constant between the two comparative cases of interest (clean vs. dust-contaminated snow). To evaluate mechanical stability, a stability indicator was derived from MEPRA, a simulation tool that is used operationally in France to assess the mechanical stability of simulated snow profiles. After establishing the framework, the authors used this model-chain to thoroughly evaluate both dry snow and wet snow conditions over a variety of aspects and elevations for their numerical study site, the Thabor Massif, which was also applied to one case-study of an actual dust-on-snow event that occurred in 2021.

**Strengths**: Although as a reviewer I am admittedly not an expert on all the inner-workings of the models mentioned above, I found the ensemble approach to handling snowpack property uncertainty and the use of a simple stability indicator from MEPRA convincing. Perhaps more importantly, I found the results of the study interesting and of broad significance to those attempting to better understand the role that light absorbing particles (LAP's) play in snowpack metamorphism for both climate-related and avalanche forecasting applications. For instance, the conclusion that LAP's deposited on the snow surface could act to both weaken and/or strengthen its mechanical stability of a dry snowpack, or that the spring wet snow avalanche cycle might occur in advanced timeframes on southern exposures, are both areas that are of relevance to operational avalanche or hydrological forecast programs. Given that there is currently a severe lack of controlled studies on this topic in the cryospheric sciences (numerical, laboratory, or field-based), to my knowledge, I applaud the authors in tackling this topic and providing a strong basis for future work in this area of research. Last, I particularly appreciated what seemed like an innovative approach to displaying the wide variety of statistical results produced, utilizing relatively easy-to-comprehend plots, elevation/aspect rosettes, and tables to present their data.

**Weaknesses**: The only weaknesses I found in the study, also acknowledged by the authors in the Discussion, was 1) that their study was limited to only one simulated and one actual dust-on- snow event, 2) that ESROC cannot resolve the vertical resolution of millimeter-scale snow processes that may occur at ice-snow interfaces, such as enhanced faceting, therefore rendering the MEPRA-derived stability indicator as somewhat limited in its regard for predicting some likely cases of snow instability, and 3) that weak layers were not tracked in such detail as to be able to be continually reassessed as additional snowfall further buried them or compared against other persistent weak layers already in the snowpack prior to the dust-on-snow event (e.g. depth hoar). This being said, given the scope of the study presented, and the abundance of data already needing to be parsed in

their analysis, it seems acceptable to me that these potential weaknesses in the study only be acknowledged, which they were by the authors in the Discussion, and then perhaps left to be revisited in subsequent studies.

In conclusion, I found no major weaknesses in the study presented, and again applaud the authors on providing the snow research community with a detailed numerical study on a topic that is of both relevance and need to a broad set potential end-users. Minor edits are given below.

Sincerely,

Kevin Hammonds
Director, Subzero Research Laboratory
Assistant Professor, Civil Engineering
Montana State University, Bozeman, MT, USA

We are really thankful to Kevin Hammons for the time dedicated to review this paper and for providing a very interesting, constructive and helpful feedback. We answer point by point below. The original review is reported in green and associated with the answers in black. Quotations from the paper are in italic font and proposed changes in purple italic font.

We fully agree with the different weaknesses identified by the reviewer.

1) The present study present two cases (described in sections 2.2.1 and 2.2.2). The synthetic dust event was selected across a larger variety of simulations because it allows for a potential impact of the dust deposition on the snowpack properties, as explained in section 2.2.1. We believe this is sufficient to illustrate the different possible impacts of dust deposition on snowpack stability (increase, decrease or unchanged dry snow stability and earlier wet snow instability). However, this is obviously a limited variety of possible cases and we discuss in the fourth paragraph of discussion: *Another limitation lies in the fact that our study is restricted to two cases [. . .].*

2) Crocus and consequently ESCROC is not designed to represent millimeter-scale processes, especially the processes that may occur at ice-snow interfaces. Moreover, we selected only one point of view on snow stability that do not take into account the most recent knowledge on stability analysis from snow profiles, with the use of MEPRA. MEPRA was designed to identify slab structures and is able, in our test case, to identify the relevant slab structures we are interested in. We discuss it in section 4.3: *We used the MEPRA stability indicator for estimation of the dry-snow stability and implemented the LWCindex for wet snow, while a large variety of other stability indicators exists (Viallon-Galinier et al., 2022) [. . .].*

3) Crocus includes an algorithm to limit the number of snow layers by merging layers that are sufficiently close in terms of internal properties [Vionnet et al., 2012]. However, merges are done based on similarity of internal properties of the layers, which means that a merge is allowed only if the layer properties are sufficiently close. This ensure that weak layers are not merged until its internal evolution make them sufficiently close to the surrounding layers. Moreover, Crocus represents the physical processes at the macroscopic scale and is therefore unable to represent millimeter-scale processes [Hammonds et al., 2015].

To summarize the last previous points, and comments of the other reviewer, we added a new paragraph in the discussion on the limits of the Crocus model:

*The snow cover model Crocus comes with some limitations. Although the mass of deposition is computed by the Crocus snow cover model, it is aggregated to the snow surface layers without modification of microstructure properties so that surface hoar can no longer be tracked as a weak layer in the model. It is not yet clear how dust deposition might affect the surface hoar formation as different processes might be involved. The presence of dust near the surface of snowpack modifies the surface temperature of snow. The dust particles may modify the condensation of ice at the surface of snow as this is the case for snow flakes in the atmosphere [Mohler et al., 2006]. Moreover, Crocus represents the physical processes at the macroscopic scale and is therefore unable to represent millimeter-scale processes [Hammonds et al., 2015].*

**Minor Edits and Clarifications:**

Page 2, Line 5: I don't see a need to use the language, "The lack of a clear message. . . ", when this is a very current area of research for multiple research groups in the cryospheric sciences.

We agree that this formulation was unnecessary. The sentence was rewritten accordingly and now reads: *The presence of a dust layer was often associated with a decrease of snowpack stability, without a clear demonstration of the link between both.*

> Page 2, Line 15: recommend replacing "dropped off" with deposited.

We changed *dropped off* to *deposited*.

> Page 2, Line 26: "albedo decrease in the visible and near-infrared". Isn't snow highly absorptive in the NIR at 1500nm? Such that dust impurities on the surface could actually increase the albedo? Perhaps clarify via the wavelengths that are being referred to here and elsewhere in the article. . . or clarify that with your model you are only using the integrated spectral albedo and over which wavelengths.

Yes, snow is highly absorptive in the NIR at 1500 nm and dust could contribute to increase the reflectance in such wavelengths (e.g. figure 4 in Dumont et al., 2020). The decrease in the near-infrared is due to the decrease in SSA due to the additional absorbed energy. Our sentence line 26 was indeed confusing and was rewritten : *The additional energy absorption accelerates the snow metamorphism and leads to a coarsening of the snow microstructure. In response to this coarsening, the snow albedo decreases in the visible and near infrared domain (300-2500 nm).*

In the methods section, we also clarified which wavelength resolution is used with TARTES and now reads:

*Second, snow cover simulations are performed with the detailed snow cover model Crocus which simulates snow physical properties by computing the mass and energy exchange within the snowpack and between the snowpack, the soil and the atmosphere [Vionnet et al., 2012]. Recent developments to represent light absorbing particles in Crocus [Tuzet et al., 2017] facilitate computing their radiative impact with the TARTES (Two-stream analytical radiative transfer in snow [Libois et al., 2013]) radiative transfer model. Note that the activation of this option is the main difference with the operational set-up described in [Morin et al., 2020]. In this study, the spectral radiative transfer scheme TARTES is used with a 20 nm spectral resolution over the range 300-2500 nm to calculate the solar energy absorbed in each 20 nm band and each snow layer. The calculation accounts for the snow microstructure as well as the quantity and type of light absorbing particles for every snow layer. It also accounts for the solar zenith angle and the spectral distribution of direct and diffuse solar radiation. More details are provided in Tuzet et al. [2017].*

> Page 3, Line 2: Grammar. . . recommend deleting this sentence and extending the previous sentence with something like ", such that we only focus in the present study. . . "

We merged the two sentences, as suggested.

> Page 3, Line 3: Replace "Up to date," with To date.

Corrected.

> Page 4, Lines 23,24, 27: Could you please expand upon what "snow physical processes" exactly are being evaluated and/or considered relevant to the remainder of the study? As opposed to just referring to them somewhat generically?

We added two sentences to precise which snow physical processes are taken into account in the construction of the ESCROC ensemble to define the modelling uncertainty: *ESCROC is an ensemble of parameterizations of the snow cover model providing estimates of the uncertainty due to the representation of the main simulated physical processes. It includes uncertainties on the properties of the new snow, snow metamorphism and compaction, liquid water percolation, and energy balance computation. The uncertainty on energy balance computation is represented through different parameterizations of turbulent fluxes at the top of the snowpack, thermal conductivity of snow layers and soil-snow exchanges.*

> Page 4, Line 28: Could you provide some examples of what meteorological forcing uncertainties are not accounted for? For instance, I'm curious about solar angle, diffuse vs. direct lighting conditions, localized heat advection in the Thabor Massif, etc.

We meant that the ensemble simulation only represented the uncertainties due to the snow model and not the one on the meteorological inputs. We provide more details in the revised version in section 2.1 of the manuscript on the effects that are accounted for:

*First, the meteorological forcing is produced by the SAFRAN meteorological analysis system. SAFRAN computes the weather conditions at hourly intervals across the French mountain ranges by analyzing meteorological surface observations from various networks [Vernay et al., 2022]. The effect of local topography on meteorological parameters, e.g. local solar masks, are not accounted for in the simulation. In the presented simulations, two types of light absorbing particles are considered: dust and black carbon. Black carbon deposition fluxes are forced by the regional climate model ALADIN-Climate which simulates the emission, the atmospheric transport and the surface deposition of black carbon [Nabat et al., 2014; Drugé, 2019]. Dust deposition fluxes are adjusted as explained in Section 2.2.*

*Second, snow cover simulations are performed with the detailed snow cover model Crocus which simulates snow physical properties by computing the mass and energy exchange within the snowpack and between the snowpack, the soil and the atmosphere [Vionnet et al., 2012]. Recent developments to represent light absorbing particles in Crocus [Tuzet et al., 2017] facilitate computing their radiative impact with the TARTES (Two-stream analytical radiative transfer in snow; Libois et al. [2013]) radiative transfer model. Note that the activation of this option is the main difference with the operational set-up described in [Morin et al., 2020]. In this study, the spectral radiative transfer scheme TARTES is used with a 20 nm spectral resolution over the range 300-2500 nm to calculate the solar energy absorbed in each 20 nm bands and each snow layer. The calculation accounts for the snow microstructure as well as the quantity and type of light absorbing particles for each snow layer. It also accounts for the solar zenith angle and the spectral distribution of direct and diffuse solar radiation. More details are provided in Tuzet et al. [2017].*

We also change the sentence line 28: *The ensemble members thus represent the uncertainty of the snow cover model but do not account for errors in the metorological input.* We hope that it is less confusing now.

> Page 5, Line 8: Similar to above comment... what are the "targeted processes" exactly? Could you be more specific, where here or elsewhere these processes could be outlined more formally?

By *targeted processes*, we mean the potential impacts of dust deposition on snowpack stability listed in previous studies: increase, decrease or no change in stability for dry snow depending on the timing of the dust deposition and earlier wet snow instability. Those are detailed in the last but one paragraph of the introduction. We reworded the sentence accordingly and now reads: *It also brings the advantage that all the potential impacts of dust deposition on snowpack stability listed in previous studies [Landry, 2014] can be studied on a single case.*

> Page 5, Line 13,14: It is not clear to me why black carbon is used as an impurity in the no-dust ensemble, could you please elaborate? Is a constant background of some black carbon in the snowpack always assumed? What concentration? Why not run the clean-snow case without any impurities?

The goal of this study is to study the impact of Saharan dust deposition of snowpack stability. To do so, the only difference between the ensembles is the presence or absence of dust. The radiative impact of dust is not the same for a clean snowpack or for a snowpack that already contains other light absorbing particles such as black carbon. This is why we included black carbon in both ensembles. As explained in section 2.1 and section 2.2, the black carbon deposition fluxes are forced with the ALADIN-Climate model that simulate the emission, the atmospheric transport and the surface deposition of black carbon.

In section 2.1, we did the following changes:

*Black carbon deposition fluxes are forced by the regional climate model ALADIN-Climate that simulate the emission, the atmospheric transport and the surface deposition of black carbon [Nabat et al., 2014; Drugé, 2019]. For both light absorbing particles, the grid point of ALADIN-Climate closest to the location is selected [Reveillet et al., 2022]. Dust deposition fluxes are adjusted as explained in Section 2.2..*

And in section 2.2.1:

*For the dust ensemble simulation, TARTES computes the spectral albedo considering both black carbon and dust, while in the no-dust ensemble the evolution of the albedo is calculated considering black carbon the only light absorbing particle present in the snowpack. Black carbon is included in both ensembles as the radiative impact of dust is not the same for a clean snowpack or for a snowpack that already contains other light absorbing particles such as black carbon. The comparison between the dust ensemble and the no-dust ensemble thus provides a numerical estimation of the impact of the dust deposition event on the evolution of snow properties as well as the associated modelling uncertainties.*

> Page 5, Line 20: Do you have any ideas about modelling uncertainties on the radiative transfer scheme? Orders of magnitude or otherwise? Potential impacts of these and other uncertainties?

All the members of the ESCROC ensemble share the same setup of the TARTES radiative transfer scheme. Hence, we do not model the uncertainties related to the energy absorption computation by TARTES. However, this radiative transfer scheme and the input data have been extensively evaluated close to Thabor massif using field measurements during two winters by Tuzet et al. [2020]. Thus, we believe that the order of magnitude of the impact is realistic for dust deposition at this location. We added a new paragraph in section 4.3 to discuss this point:

*Modelling errors might also affect the radiative transfer scheme. All the members of the ESCROC ensemble use the same setup of the TARTES scheme. This radiative transfer scheme uses simplifying assumptions for light absorbing particles: the particles are considered as Rayleigh scatterers and the model does not account for the position of the particles with respect to the ice matrix [Hagenmuller et al., 2019]. This might influence the estimated radiative impact. However, the radiative transfer scheme and the input data have been extensively evaluated close to the Thabor region using field measurements during two winters [Tuzet et al., 2020]. Thus, we believe that the order of magnitude of the impact is realistic for dust deposition at this location.*

> Page 6, Line 18: Could you please list all the snow mechanical properties that are output from MEPRA? As opposed to "e.g., shear strength or ram resistance"...is it one or the other or both? More?

We now provide a more complete and precise description of MEPRA: *The model MEPRA analyses at every time step the output of the snow cover model, calculates the snow mechanical properties of the snow layers (shear strength and rammsonde penetration resistance), performs a basic stability analysis (shear strength-stress ratio at each layer boundary, with or without skier [P. Fohn, 1987]) and based on that data, provides stability indicators for the probability of natural release and artificial triggering.*

> Page 7, Line 20: Delete ""we" from "member we the number"

Corrected.

> Page 7, Line 30: Please provide a citation for 0.03 kg per m^3.

The numerical value is the one proposed by Mitterer et al. [2013]. We adapted the sentence to make clear that the definition, including the numerical threshold selected, is the original one from Mitterer et al [2013]: *In this study we use the liquid water content index ($LWC_{index}$) as introduced by Mitterer et al. [2013]. The $LWC_{index}$ is calculated in this study by dividing the mean volumetric liquid water content of the simulated snowpack by a typical value of 0.03 kg $m^{-3}$.*

> Page 10, Line 1: Delete "of" from "of of"
>
> Page 22, Line 7: Delete "the one" from "than the one our"
>
> Page 23, Line 1: Add "a case" to "case"
>
> Page 23, Line 25: Add an s to "eruption", to make the plural "eruptions"
>
> Page 23, Line 26: Delete "es" from "ashes".

We corrected the typos.

---

## Author Comment (AC2)

**Answer to RC2 (Ingrid Reiweger)**

Oscar Dick    Léo Viallon-Galinier    François Tuzet    Pascal Hagenmuller
Mathieu Fructus    Benjamin Reuter    Matthieu Lafaysse    Marie Dumont

**General comment**

The paper "Can Saharan dust deposition impact snowpack stability in the French Alps¿' deals with the question on how Saharan dust deposition influence snowpack stability with respect to avalanche release. To investigate the subject, the authors use both real and simulated snowpacks with and without dust and calculate stability indices for both snowpacks (dust vs. no dust). These stability indices are then compared to estimate the dust's influence on snowpack stability. The short answer to the initial question is, not surprisingly, found to be "it depends". Nevertheless, the paper provides interesting insights, cool modelling chains with ensemble modelling, as well as splendid presentation of the resulting statistics, and I would certainly recommend publication, taking into consideration the detailed comments below.

We are really thankful to Ingrid Reiweger for providing very useful feedback on this paper. We answer, in black, to the detailed comments below. Quotations from the paper are in italic font and proposed changes in purple italic font.

**Detailed comments**

2.3 Here suddenly you change to present tense!

p.6 l. 24 are not

Corrected.

p.6 l. 26 The stresses due to a skier or due to the load of the overlying snow layers?

We now precise in the text that both contributions are taken into account: *Level 3, the highest level, on the other hand, corresponds to pronounced instability where the stress in this layer, due to the weight of overlying layers and additional load, is close to layer strength.*

p.23 l. 13: had not

Corrected.

Formula 1: I guess i = 1 ... i_max What is i_max?

We now precise in the introductory sentence of this equation that we consider 35 members.

p.6. l.28 What about surface hoar as a weak layer?

Moreover, I really wonder whether dust inhibits growth of surface hoar and this increases snowpack stability for this particular case?

Crocus snow cover model is not able to represent surface hoar (SH) layers. Although the mass deposition is represented, the model is not designed to track these layers and not have a specific representation of surface hoar as SNOWPACK model have. SH layers are a significant concern in many regions [e.g. Hageli et al., 2013], and reviewer's question is therefore highly relevant. The interaction of surface hoar formation and evolution with dusts largely remains to be studied. We added a paragraph in the discussion to underline this limitation:

*The snow cover model Crocus comes with some limitations. Although the mass of deposition is computed by the Crocus snow cover model, it is aggregated to the snow surface layers without modification of microstructure properties so that surface hoar can no longer be tracked as a weak layer in the model. It is not yet clear how dust deposition*

*might affect the surface hoar formation as different processes might be involved. The presence of dust near the surface of snowpack modifies the surface temperature of snow. The dust particles may modify the condensation of ice at the surface of snow as this is the case for snow flakes in the atmosphere [Mohler et al., 2006].*